# Exploring the Spatial Effects of Built Environment on Quality of Life Related Transportation by Integrating GIS and Deep Learning Approaches

Pawinee Iamtrakul [1,*], Sararad Chayphong [1], Pittipol Kantavat [2], Yoshitsugu Hayashi [3], Boonserm Kijsirikul [2] and Yuji Iwahori [4]

1   Center of Excellence in Urban Mobility Research and Innovation, Faculty of Architecture and Planning, Thammasat University, Pathumthani 12120, Thailand
2   Department of Computer Engineering, Faculty of Engineering, Chulalongkorn University, Bangkok 10330, Thailand
3   Center for Sustainable Development and Global Smart City, Chubu University, Kasugai 487-8501, Japan
4   Department of Computer Science, Chubu University, Kasugai 487-8501, Japan
*   Correspondence: pawinee@ap.tu.ac.th

**Abstract:** Understanding the quality of life related to transportation plays a crucial role in enhancing commuters' quality of life, particularly in daily trips. This study explores the spatial effects of built environment on quality of life related to transportation (QoLT) through the combination of GIS application and deep learning based on a questionnaire survey by focusing on a case study in Sukhumvit district, Bangkok, Thailand. The Geographic Information System (GIS) was applied for spatial analysis and visualization among all variables through a grid cell ($500 \times 500$ sq.m.). In regard to deep learning, the semantic segmentation process that the model used in this research was OCRNet, and the selected backbone was HRNet_W48. A quality-of-life-related transportation indicator (life satisfaction) was implemented through 500 face-to-face interviews and the data were collected by a questionnaire survey. Then, multinomial regression analysis was performed to demonstrate the significant in positive and negative aspects of independent variables (built environment) with QoLT variables at a 0.05 level of statistical significance. The results revealed the individuals' satisfaction from a diverse group of people in distinct areas or environments who consequently perceived QoLT differently. Built environmental factors were gathered by application of GIS and deep learning, which provided a number of data sets to describe the clusters of physical scene characteristics related to QoLT. The perception of commuters could be translated to different clusters of the physical attributes through the indicated satisfaction level of QoLT. The findings are consistent with the physical characteristics of each typological site context, allowing for an understanding of differences in accessibility to transport systems, including safety and cost of transport. In conclusion, these findings highlight essential aspects of urban planning and transport systems that must consider discrepancies of physical characteristics in terms of social and economic needs from a holistic viewpoint. A better understanding of QoLT adds important value for transportation development to balance the social, economic, and environmental levels toward sustainable futures.

**Keywords:** central business district (CBD); life satisfaction; semantic segmentation process; sustainable transportation; well-being





## 1. Introduction

Quality of life (QoL) emerged in the 1980s as a popular term to describe cities [1] and has been operationalized in various ways [2,3], which has presented a multifaceted concept used by a variety of disciplines. Based on the understanding most people have of goodness of life, happiness, living successfully, or life satisfaction, a holistic model is essential for

understanding multidimensional components related to the evaluation of passengers' journeys [4–6]. In addition, quality of life (QoL) requires a complex measurement due to several influencing factors. A number of studies provide a classification concept of the contributing factors associated with QoL, and the categorization can be classified into four classes to explain an individual's experience on their journey. The range of variety of description of each class includes an objective evaluation, subjective measurement, a combination of objective and subjective measurement, and domain-specific measurement (e.g., friendships, leisure, housing, transportation) [7,8]. This study focuses on the domain-specific quality of life in transportation (QoLT) since transportation usually involves individual residents and commuters in a multi-dimensional construct, which is critical to people's lives. It also can be referred to as the individual's experience of life, which is related to the evaluation of their lives. Furthermore, it can contribute to delivering an evaluation of differences in experiences among a variety of modes of transport from the commuter's viewpoint.

Nonetheless, as the city has grown, the intensity of services and activities within the city has also increased, including the growth in transport demand and transport developments, especially the development of road networks. With the high employment opportunities and high population density, it has resulted in heavy traffic of commuters to the inner area of the capital city. In such a situation, the central business district presents major attractors involving professional jobs mixed with the built environment for other activities (e.g., business, commercial, leisure and recreation, etc.). More engagement with these frequent activities, and therefore more time spent on the road by urban commuters, could have a direct effect on their QoLT, and the attributes of the built environment could have a significant impact on the satisfaction of their journey. Although expanding the road network improves travel connectivity, it facilitates additional private vehicle usage, leading to congestion problems, increasing vehicle emissions, and degrading ambient air quality. Besides, transport impacts people's quality of life tremendously via the externalities of the current system, such as traffic accidents, poor accessibility, impacts on mental health, etc. [9]. The abovementioned problems indicate that the amount of evidence correlating transportation with quality of life, and therefore understanding people's lives in connection with land use and transportation, remains small. Consequently, with a comprehensive understanding, transport improvements can promote a better quality of life, which requires a recommendation to ensure all residents and commuters have access to good transport options for safe, healthy, affordable, and accessible transportation [10,11]. From a transportation perspective, interest in QoL has been increasing in several studies worldwide. Yoshitsugu and Ikuo (2003) introduced a method to evaluate individual inhabitants' QoL based on accessibility to locations of service facilities such as hospitals and shops [12]. Furthermore, Achariyaviriya et al. (2021) also recommended a method to evaluate the effects of shifting workplace locations and daily commuting periods based on QoL gained from alternative sequences of travel and activities in a day [13]. For instance, Mattson et al. (2021) examined the impact of a community's transportation system on the quality of life provided to its residents [14]. It indicated that walkability and street characteristics are associated with community quality of life. Likewise, Spinney et al. (2009) found a significant association between transportation mobility benefits and quality of life [15]. Overall, transportation is a likely contributor to access to services and activities, especially necessary physical and social activities/services [10,16], which leads to many opportunities for access to activities. Evidence indicating life satisfaction refers to a better quality of life and well-being [17,18].

Transportation-related quality of life (QoLT) is measured by multiple indicators, which are associated with physical, economic, mental, and social well-being [7,19]. However, the WHO developed an instrument called WHOQOL-BREF, which represents an assessment based on an individual's perception. Thus, at the individual level, there is no definitive approach to understanding individual position in the multi-context of culture, society, level of economic development, and environment in which people's lives are related to disparate perceptions [20,21]. The viewpoint of individuals, either passengers or drivers

from distinct areas or environments, have perceived quality-of-life differences. This multi-faceted issue has made studying more challenging and stimulating. The issue concerning a variety of people in different areas and unique environments who perceive quality of life differently consequently means that the urban environment plays a key role as one of the indicators for the measurement of quality of life. The terms of environment, which can be wide-ranging from the natural environment (greenspaces, lightly populated) to the built or physical environment (man-made, densely populated) to the social environment (family, peers, community engagement), serves as the context of life and contributes to quality of life. Regarding the built or physical environment, it can be considered a component of the spatial layout for describing all physical resources built by humans (e.g., buildings, structures, and facilities). Several studies have demonstrated the association between the built environment and quality of life; for instance, Cerletti et al. (2021) confirmed that the built environment plays a crucial role in health-related quality of life [22]. Chaudhury and Xu (2022) focused on the built environment's characteristics and influence on residents' quality of life in long-term care facilities [23].

It is well known that several studies have pointed to quality of life being directly influenced by social–economic status. However, fewer studies have revealed an association between the built environment and transportation-related quality of life (QoLT). Furthermore, with the emergence of new tools, e.g., smartphones, IoT technologies, and artificial intelligence, for instance, Zhao et al. (2021) studied the impact of data processing on deriving micro-mobility patterns from available vehicle data [24], and the study of Sun et al. (2019) focused on the application of data processing to derive mobility patterns from passively generated mobile-phone data [25]. However, the intangible value of quality of life in term of the social dimension is one of the most important issues that has been studied for a long time and still needs to be understood and studied widely, since it helps gain a comprehensive understanding of most people's goodness of life, happiness or successful living, and life satisfaction. Thus, several studies have mainly applied satisfaction questionnaires for collecting perception on an individual level to represent their reflections on their daily living. Therefore, the integration of techniques in this consideration is an important issue in order to collect both spatial interpretation and perceptual data. Besides, several studies have mainly applied questionnaires or physical survey approaches in built environment-related spatial dimensions.

Additionally, artificial intelligence (AI) technologies have also been popular for generating urban big data to support urban design and urban planning. AI has been developed very promptly recently, and there are several studies applying this tool for gathering data and manipulating data for quality-of-life evaluation. Moreover, the spatial approach is widely used for assessment by employing the Geographic Information System (GIS) (Haider and Iamtrakul, 2022), which presents a robust spatial-approach tool in quality-of-life evaluation and visualized demonstration [26]. For instance, Olajuyigbe et al. (2013) adopted the GIS analytical tool to assess quality of life [27]. Furthermore, Amin et al. (2021) also performed a spatial study of quality of life in the USA by applying spatial data and to create maps with significant clusters based on GIS-based mapping [28]. The intersection of artificial intelligence (AI) and GIS allows for massive opportunities for exploring the spatial dimensions of a multisource approach to quality-of-life determination. Regarding knowledge extraction by using deep-learning techniques, the image-recognition techniques that are frequently used are semantic segmentation and object detection, and some studies have adopted these techniques for the QoLT-related context, including Kantavat et al. (2019), Fukui et al. (2022), and Thitisiriwech et al. (2022) [29,30]. Therefore, this study aims to explore the spatial effects of built environments on quality-of-life-related transportation based on an integration of GIS and deep-learning techniques. In the remainder of this paper, the literature review (Section 2) of previous studies is summarized to position the QoL and related concepts that are adopted in this paper. In Section 3, the methodology, the approaches for gathering the data, and the method of analysis are explained together with a description of the study area. In Section 4, the descriptive analysis together with a

multinomial logistic model is applied to provide better understanding about the different levels of QoLT based on model estimation. Furthermore, the clustering model is employed to understand the different urban context related to the level of variation of QoLT. Finally, the summary of the results of the analysis is presented and policy recommendations are discussed in the last sections (Sections 5 and 6, respectively).

## 2. Literature Review

Quality of life (QoL) can be classified as an objective domain [31], a subjective domain (Diener, 2000) [5], a combination of objective and subjective [32], and domain specific [33,34]. Transportation is a domain-specific aspect of quality of life. The concept of QoL is applied in transport studies due to its purpose of better understanding and evaluating commuters' satisfaction during their journey. Rather than measuring one or two components, it allows for the evaluation of multidimensional views in one holistic model. This research reviewed the critical variables related to QoLT and summarized them as elaborated in Table 1. Furthermore, the built environment related to QoLT can be demonstrated in the relationship shown in Figure 1.

**Table 1.** Variables.

| Main Variable | Sub-Variable | Details | References |
|---|---|---|---|
| Quality of Life | | | |
| Quality-of-life-related transportation (life satisfaction) | Accessibility | Access to destinations or people's ability to reach the destinations in order to meet their needs and desire to visit or to satisfy their wants | [7,35–37] |
| | Design | Describes the physical layout of the transportation system and includes the multiple components of the system (e.g., roads, signs, and lights) | [38,39] |
| | Safety | Refers to a problem caused by transportation (e.g., traffic accident) | [14,40–42] |
| | Cost | Describes the affordability of the transportation system (e.g., travel costs) | [36,43,44] |
| | Environment | Refer to the externalities or impacts caused by transportation (e.g., noise, air pollution) | [39,45,46] |
| | Mobility | Describes the experience involved with the movement of people from the origin to the destination related to daily life | [10,15,47] |
| | Information | Refers to information and communication of data in transportation (e.g., signage, traffic signal) | [43] |
| Built Environment | | | |
| Road components | Semantic segmentation of road scenes (e.g., road, sidewalk, building, wall, fence, pole, traffic light, traffic sign, vegetation, terrain) | Percentage of observed areas (in pixels) in image. | [29,30,48] |
| Land use | Land-use characteristics | Characteristics of land use that refer to urban density, diversity of activities in the area (e.g., residential, commercial, mixed-use) | [7] |
| Mode of transportation | Modes | Mode choices in transportation systems (e.g., active transportation, public transportation, paratransit) | [7] |

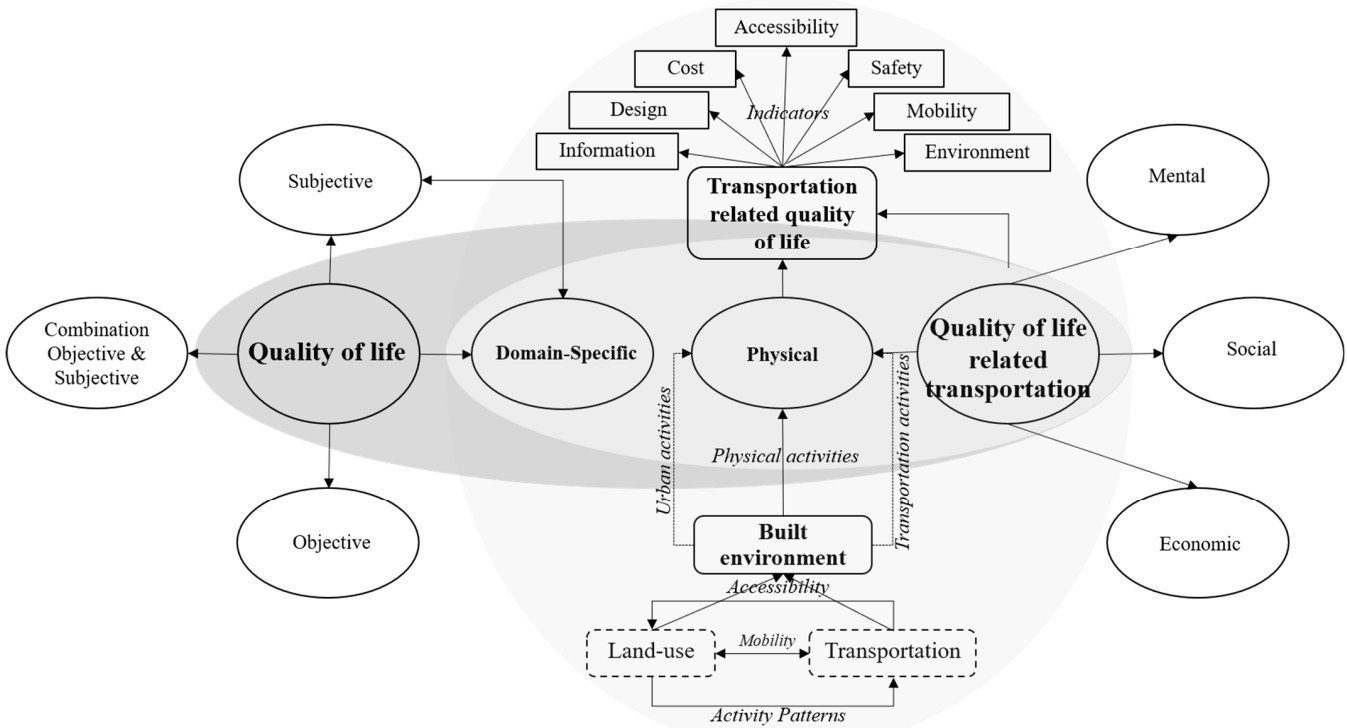

**Figure 1.** Conceptual framework.

### 2.1. Built Environment and Quality of Life Related to Transportation (QoLT)

Understanding the quality of life related to transportation is one of the key issues to allowing a comprehensive assessment of the differences in commuters' quality of life. The built environment plays a role as one of the measurement indicators of quality of life due to the significant features of the environment related to individuals' perceptions about quality of life. After considering previous studies about transportation-related quality of life (QoLT), it is clear that there are many indicators related to this evaluation, including mobility [15,47], affordability/cost [36,44], safety [40,41], accessibility [7,36,37], connectivity [49], environmental impact [39,45], design [38], etc. In terms of measurement for built or physical environment related to QoLT, attempts have been made to assess transportation-related physical activities [50,51], e.g., mixed land use, residential density, street connectivity, etc. [52]. Likewise, Lee and Sener (2016) defined indicators related to the built environment as active travel/transit, land use, accidents/safety, and walkability [7].

### 2.2. Quality of Life Related to Transportation (QoLT) and Sustainable Development

Sustainable city development is a development concept that aims at ensuring better quality of life for everyone based on stimulating economic, social, and environmental development. The United Nations Development Programme (2022) demonstrated the concept of the Sustainable Development Goals (SDGs), which is the primary goal of development related to urban development and quality of life [53]. Thus, the key to development is focused on three components (social, economic, and environmental) to ensure an approach to sustainable development for a better quality of life for all groups of people in the diverse context of city/community. Regarding sustainable development-related transportation or sustainable transportation, transportation plays an essential role as a critical element when studying quality of life. Transportation provides a platform to meet daily life demands and opportunities in people's lives in various activities, e.g., economic, leisure and recreation, etc. [16].

*2.3. Integrating Deep Learning for Assessing Quality of Life*

To understand the physical attributes of urban areas, data can be gathered by several sources, particularly by integrating technology advancement, i.e., artificial intelligence (AI) techniques. Nowadays, machine learning (ML), a subfield of artificial intelligence (AI), has continuously evolved and been applied to various applications. Deep learning (DL) is one of the most powerful ML techniques and has been adopted in the problem domain. DL consists of an input layer, hidden layers, and an output layer.

The existing literature on machine- and deep-learning methods demonstrates their development, and applications have been employed over the past decade. Regarding information extraction from images, this research applies the image-recognition technique of semantic segmentation. Semantic segmentation, also called image segmentation, is the process of clustering or defining the object–class sections of an image in the form of pixel-level prediction. Each pixel section of an image is classified into a category or class. Some examples of datasets widely used for this task are Cityscapes [54] and PASCAL VOC [55]. The most closely related dataset in this research is Cityscapes, a dataset that is considered a state-of-the-art, high-quality, and significant variation benchmarking suite. The dataset consists of images from more than 50 cities in Europe with various street scenes, mainly captured in Germany.

Semantic segmentation of road scenes deep learning is a blooming field, and in the rise of deep-learning technologies, convolutional neural networks are applied to image segmentation. However, neural networks have several series for handling semantic-segmentation problems—dual-resolution networks with deep high-resolution representation. Semantic segmentation presents a crucial technology for understanding the surrounding scenes [56], in which the search pixel of the input image should be assigned to the corresponding label [57,58]. This research used HRNetV2 + OCR [25,59]. The model starts from high-resolution-image convolution neural networks (CNNs) and then gradually decreases the resolution of the convolution but still uses the previous higher resolution in parallel. The convolutional neural network (CNN) (Krizhevsky et al., 2012) is a deep-learning (DL) model for computer vision [60]. The architecture of CNN was first proposed by Fukushima (1988) [61] and was vitally improved by introducing a gradient-based learning algorithm by Lecun et al. (1998) [62]. Since that time, there has been a lot of research that has improved CNN models. Currently, CNN is widely accepted as a standard method in machine learning for image processing [63]. Compared to the traditional deep-learning (DL) model that fully connects all nodes between layers using edges, CNN prunes the edges using two mechanisms: convolutional layers and subsampling (or max-pooling) layers.

## 3. Methodology

This study aimed to explore the spatial effects of built environments on quality-of-life-related transportation by integrating a questionnaire survey to collect subjective data based on life satisfaction among six indicators (accessibility, design, safety, cost, environment, mobility, information). The Geographic Information System (GIS) was used to collect and visualize all variables of the built environment through a grid cell (500 × 500 sq.m.). Furthermore, the deep-learning technique was used to collect road and physical components, which is one of the built environments, by applying the semantic-segmentation process of the HRNetV2 + OCR model. Finally, the exploration of the spatial effects between built environments and quality-of-life-related transportation via statistical analysis is explained in detail below.

*3.1. Data Collection and Study Area*

Sukhumvit district, Bangkok, Thailand, is presented as a case study and consists of three districts (Vadhana district, Khlong Toei district, and Phra Khanong district), as depicted in Figure 2. Sukhumvit district, Bangkok, is presented as one of the central business districts (CBD). The area is considerable, with information at the bottom indicating the best data sources, including a collection of facilities such as those with programming and output

formats (e.g., urban components, infrastructure facilities, transportation modes, etc.). With the geography of its location, the role of employment functions and the provision of basic needs of urban residents allows people living in this area to have several choices of infrastructure and transportation, particularly in terms of travel options. The physical condition of urban geography and economic agglomerations to cope with rapid urbanization have resulted in high growth and concentration of spatial development in urban structures. Due to this area having long been a center of economic and social activities, the functions of CBD-dominated urban development has led to a high level of urban services and multiple modes of transportations to access shopping and entertainment areas, residences, and multi-stage office buildings. It is also represented as a source of high cost of living due to its location in the heart of the city. It can be considered a premium lifestyle area with various premium elements, including shops, restaurants, and services, and, most importantly, high land prices. However, the variety of activities to attract commuters and dwellers living in the area has resulted in a variety of residential choices ranging from high-rise housing to informal residential areas. Thus, the spatial spread of urban settlement has an impact on traffic jams and environmental pollution in residential and working areas. The diverse nature of the area has led to a mixture of quality of facilities, housing, and travel patterns. This area is considered a significant representative of urban problems that exert a certain degree of pressure on their inhabitants to search for a certain quality of life based on the diversity of the spatial urban and facility characteristics.

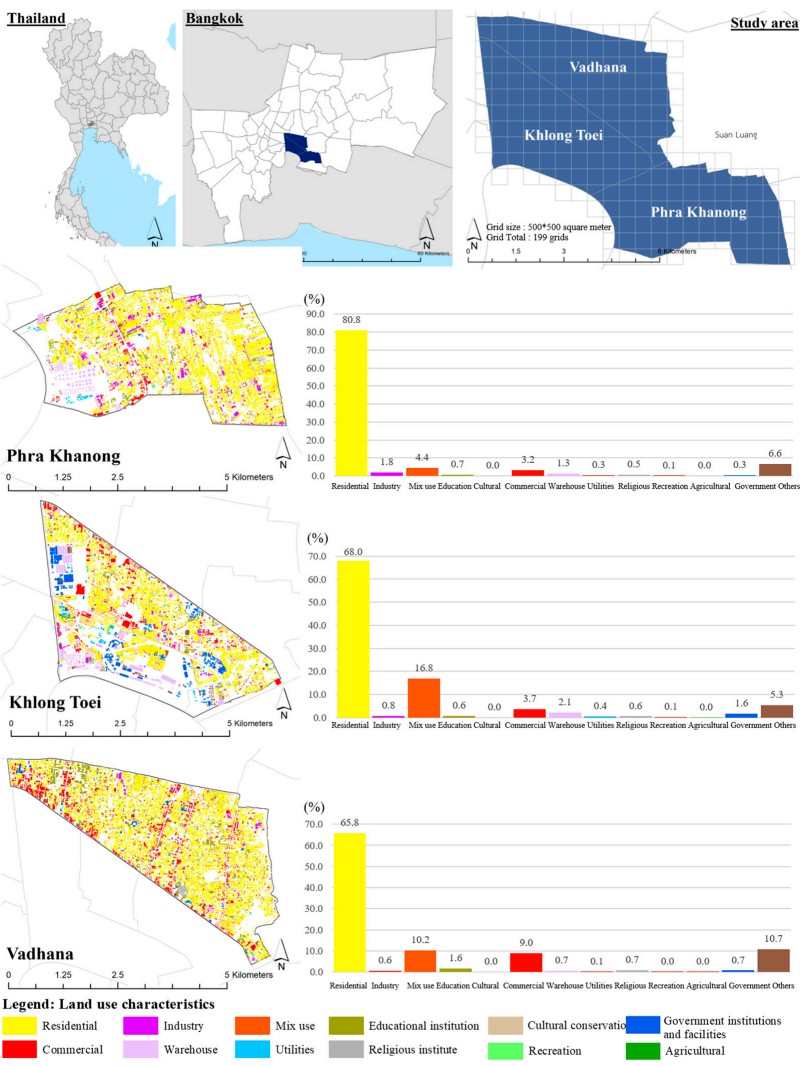

**Figure 2.** Study area: Sukhumvit district, Bangkok, Thailand.

There are several factors to be considered including physical factors and quality-of-life-related transportation (life satisfaction) factors, each of which has different analytical units. Therefore, in this study, relevant factors were prepared for a grid-based analysis. For spatial-data preparation, this study applied the grid approach, which helped delimit different areas and facilitate the utilization of a combined dataset of built environments and quality of life relating to transportation (QoLT). Several studies related to spatial-based analysis in terms of transport and built-environment dimensions have applied unit analysis extensively. Grid-based structures are one of the most popular techniques applied to spatial distribution), and demonstrate the grid size related to the size of the study area and the desired accuracy and resolution [64,65]. For instance, Smith (2011) used a 500 m × 500 m square grid to illustrate the density [66]. Likewise, Mao et al. (2020) performed the integration of big-data mining and analytics techniques with 500 × 500 m grids applied to spatial distribution of urban built-environment stock [67]. A number of studies have deployed different grid sizes; however, after reviewing the studies on the deployment of displays via grids at the area level, most of the grid sizes were in the range of 100–500 square meters. In the study, the grid-setting technique of 500 × 500 square meters was adopted for further analysis.

The data input in this study can be categorized into two components, which include the following details. The framework of study is depicted in Figure 3.

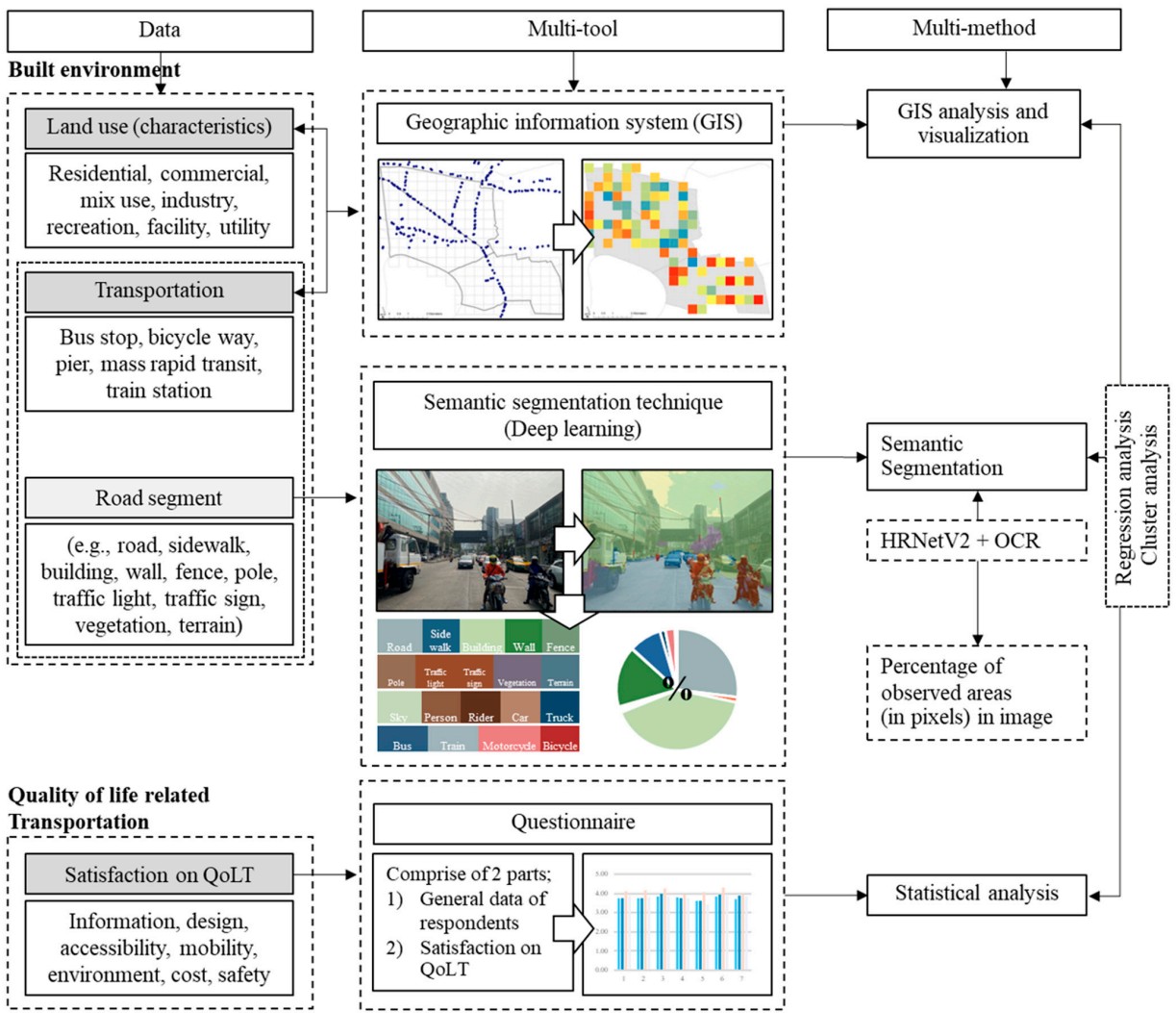

**Figure 3.** Framework of the study.

*Questionnaire surveys*: The questionnaire survey was designed based on a stratified random sample of adults aged 18 and over. It was designed to gather the data by including only respondents who had experienced travel in various modes (e.g., private automobile, active transportation, public transportation, paratransit, etc.). The questionnaire received ethical approval from The Human Research Ethics Committee of Thammasat University No. 2 Social Sciences (approval number 146/2021). Before enrollment, explanations of the details of the questionnaire together with the purpose of the data gathering were provided to the respondents, and confirmed consent was received from all data providers. Data of respondents who declined to participate until the end of the interview and incomplete data were excluded.

The questionnaire surveys described QoLT satisfaction indicators comprising information regarding design, accessibility, mobility, environment, cost, and safety. Respondents were asked to indicate the extent to which various indicators were perceived during their trips. In addition to asking questions, data on photographing road elements were also collected when the questionnaire was filled out by the respondents. The questionnaire was distributed through the grid by covering the study area across different transportation modes. The aforementioned data-collection approach allowed the QoLT indicators collected from the subjects in the grid to be included in the grid map, expressed as the average value of satisfaction in each grid. Photographic data of road elements were also collected in each grid.

*Built environment*: Data sources came from two approaches, which consisted of:

(1) *Geographic Information System (GIS) data*: comprises land-use (residential, commercial, mixed use, industry, recreation, facility, utility) and transportation (bus stop, bicycle path, pier, rapid mass transit, train station) data that were gathered from open databases and present via a grid cell.

(2) *Data extracted by using the semantic-segmentation technique:* comprises road segments (e.g., road, sidewalk, building, wall, fence, pole, traffic light, traffic sign, vegetation, terrain). This data-gathering process was extracted from photographs of road components collected via physical survey that were taken in accordance with the location of the questionnaires collected in each set. This procedure was carried out in order to provide information on the opinions of individuals in line with their travel environments.

### 3.2. Measurement and Analysis

This research applied the open-source software package "PaggleSeg" [68] in the semantic-segmentation process, which was published by PaddlePaddle Contributors (2019) [69]. The model used in this research was OCRNet, and the selected backbone was HRNet_W48, the only super parameter of which is "backbone = HRNet_W48." The model was implemented in Python and trained using the Cityscape dataset. The package includes the pre-trained model; hence, the software was ready to use, and there was no need to re-train the model.

For the measurement of QoLT, this study adopted indicators for requesting respondents to indicate their satisfaction based on the widely used World Health Organization scale. The general satisfaction with QoL was measured with the question of "How satisfied are you with the quality of your life?" using a seven-point scale, where 1 = very dissatisfied and 7 = very satisfied [21,70]. However, there are some studies that measured QoL by asking similar assessment questions and using a five-point scale for the answers [71–73]. For data analysis, statistical analyses were applied using descriptive statistics and multinomial regression analysis through the application of SPSS statistics (version 28.0). Firstly, descriptive analysis of the socio-economic status among respondents was investigated with descriptive statistics. Second, satisfaction with QoLT was compared across the different travel modes of the respondents using chi-square tests. Thirdly, the relationship between the built environment and satisfaction with QoLT was evaluated by multinomial regression analysis. Finally, built-environment variables were assessed by performing cluster analysis based on the consideration of various characteristics of physical data. Consequently, the

determination of satisfaction with QoLT was input into the next step of identification of differences in clustering associations within the study area. Among the different clusters of built-environment features, satisfaction on QoLT was differentiated using chi-square in regard to characteristics of urban patterns. All statistical tests were evaluated at a 0.05 level of statistical significance.

## 4. Results

The built environment can also affect QoLT through individuals' choice of travel mode, and accordingly, the travel mode can affect travel satisfaction through its impact on the quality of a place. This linkage is also related to destination accessibility due to the impacts on travel behavior in terms of meeting needs, providing services, and offering activities that can create better opportunities for an individual's life.

### 4.1. Socio-Economic Profile of Participants

Sukhumvit district, Bangkok, Thailand, is a vital area for business and offices, and is surrounded by commercial and economic agglomeration. This includes the expansion of new housing estates next to the evaluated train station, which results in easy access by traveling within the area. Although there are many choices of transportation modes available, e.g., rapid mass transit, bus, and paratransit, as described in Table 2, the primary mode of transportation of commuters is private cars, reflecting the problem of traffic congestion in the study area due to dependency on automobiles.

**Table 2.** Respondent profile.

| Variables | Private Automobile | | | | Active Transportation | | Public Transit | | | | Paratransit | | Total | |
|---|---|---|---|---|---|---|---|---|---|---|---|---|---|---|
| | Passenger Car | | Motorcycle | | | | Bus | | Mass Rapid Transit | | | | | |
| | n | % | n | % | n | % | n | % | n | % | n | % | n | % |
| Social Aspect | | | | | | | | | | | | | | |
| Gender | | | | | | | | | | | | | | |
| Male | 65 | 67 | 62 | 70.5 | 37 | 48.7 | 29 | 37.2 | 40 | 50.6 | 38 | 46.3 | 271 | 54.2 |
| Female | 32 | 33 | 26 | 29.5 | 37 | 48.7 | 45 | 57.7 | 33 | 41.8 | 41 | 50 | 214 | 42.8 |
| Others | 0 | 0 | 0 | 0 | 2 | 2.6 | 4 | 5.1 | 6 | 7.6 | 3 | 3.7 | 15 | 3 |
| Age (years) | | | | | | | | | | | | | | |
| 18–25 | 8 | 8.2 | 7 | 8 | 16 | 21.1 | 16 | 20.5 | 29 | 36.7 | 30 | 36.6 | 106 | 21.2 |
| 26–35 | 46 | 47.4 | 42 | 47.7 | 34 | 44.7 | 28 | 35.9 | 35 | 44.3 | 38 | 46.3 | 223 | 44.6 |
| 36–59 | 36 | 37.1 | 37 | 42 | 22 | 28.9 | 29 | 37.2 | 15 | 19 | 13 | 15.9 | 152 | 30.4 |
| Over 59 | 7 | 7.2 | 2 | 2.3 | 4 | 5.3 | 5 | 6.4 | 0 | 0 | 1 | 1.2 | 19 | 3.8 |
| Religions | | | | | | | | | | | | | | |
| Buddhist | 93 | 95.9 | 76 | 86.4 | 68 | 89.5 | 63 | 80.8 | 64 | 81 | 66 | 80.5 | 430 | 86 |
| Christian | 2 | 2.1 | 9 | 10.2 | 5 | 6.6 | 10 | 12.8 | 9 | 11.4 | 8 | 9.8 | 43 | 8.6 |
| Islam | 1 | 1 | 2 | 2.3 | 3 | 3.9 | 5 | 6.4 | 3 | 3.8 | 2 | 2.4 | 16 | 3.2 |
| Others | 1 | 1 | 1 | 1.1 | 0 | 0 | 0 | 0 | 3 | 3.8 | 6 | 7.3 | 11 | 2.2 |
| Marital status | | | | | | | | | | | | | | |
| Married | 45 | 46.4 | 45 | 51.1 | 47 | 61.8 | 42 | 53.8 | 47 | 59.5 | 54 | 65.9 | 280 | 56 |
| Single | 48 | 49.5 | 36 | 40.9 | 22 | 28.9 | 27 | 34.6 | 26 | 32.9 | 24 | 29.3 | 183 | 36.6 |
| Divorce | 3 | 3.1 | 6 | 6.8 | 8 | 7.9 | 7 | 9 | 3 | 3.8 | 2 | 2.4 | 27 | 5.4 |
| Others | 1 | 1 | 1 | 1.1 | 1 | 1.3 | 2 | 2.6 | 3 | 3.8 | 2 | 2.4 | 10 | 2 |

**Table 2.** *Cont.*

| Variables | Private Automobile | | | | Active Transportation | | Public Transit | | | | Paratransit | | Total | |
|---|---|---|---|---|---|---|---|---|---|---|---|---|---|---|
| | Passenger Car | | Motorcycle | | | | Bus | | Mass Rapid Transit | | | | | |
| | n | % | n | % | n | % | n | % | n | % | n | % | n | % |
| Economic Aspect | | | | | | | | | | | | | | |
| Education level | | | | | | | | | | | | | | |
| Junior high school | 5 | 5.2 | 3 | 3.4 | 5 | 6.6 | 1 | 1.3 | 0 | 0.0 | 3 | 3.7 | 17 | 3.4 |
| High school | 4 | 4.1 | 10 | 11.4 | 13 | 17.1 | 12 | 15.4 | 8 | 10.1 | 13 | 15.9 | 60 | 12.0 |
| Vocational college | 8 | 8.2 | 20 | 22.7 | 11 | 14.5 | 22 | 28.2 | 10 | 12.7 | 6 | 7.3 | 77 | 15.4 |
| Bachelor's degree | 67 | 69.1 | 48 | 54.5 | 44 | 57.9 | 41 | 52.6 | 55 | 69.6 | 49 | 59.8 | 304 | 60.8 |
| Postgraduate | 13 | 13.4 | 7 | 8.0 | 3 | 3.9 | 2 | 2.6 | 6 | 7.6 | 11 | 13.4 | 42 | 8.4 |
| Income Level (per person per month, THB) | | | | | | | | | | | | | | |
| Less than 10,000 | 5 | 5.2 | 3 | 3.4 | 9 | 11.8 | 5 | 6.4 | 5 | 6.3 | 10 | 12.2 | 37 | 7.4 |
| 10,001–25,000 | 34 | 35.1 | 52 | 59.1 | 31 | 40.8 | 41 | 52.6 | 34 | 43.0 | 29 | 35.4 | 221 | 44.2 |
| 25,001–40,000 | 33 | 34.0 | 17 | 19.3 | 23 | 30.3 | 22 | 28.2 | 34 | 43.0 | 22 | 26.8 | 151 | 30.2 |
| 40,001–55,000 | 14 | 14.4 | 11 | 12.5 | 9 | 11.8 | 8 | 10.3 | 4 | 5.1 | 14 | 17.1 | 60 | 12.0 |
| More than 55,000 | 11 | 11.3 | 5 | 5.7 | 4 | 5.3 | 2 | 2.6 | 2 | 2.5 | 7 | 8.5 | 31 | 6.2 |

Remark: Total data includes 500 sets.

When considering the general information of the participants, out of a total of 500 respondents, 54.2% were male and 42.8% were female. The age of the participants ranged from 18 to 60 years old (M = 33.16, SD = 9.06). There was a total of 44.6% adults (26–35 years), followed by middle aged individuals (36–59 years) and young adults (18–25 years). Buddhism was the religion of most participants (86.0%), followed by Christianity (8.6%) and Islam (3.2%). A total of 56.0% of respondents were married, followed by single participants (36.6%), divorced participants (5.4%), and others (2.0%). The data collection based on the social aspect revealed that the respondents fell into the category of working age due to the characteristics of employment and activities in the area, such as offices and business buildings. Regarding the economic aspect, most participants' education level was at a bachelor's-degree level (60.8%), followed by vocational college (15.4%). The participants' average income (THB per month per person) was about THB 10,000–40,000 per month, and 44.2% earned THB 10,000–25,000 per month. However, 7.4% were the group of earning less than THB 10,000 per month, showing that participants were of diverse socio-economic status. Although the Sukhumvit area represents an important business and commercial district, the sampling group of residents and commuters within the site earned relatively low incomes compared to the relatively high cost of living in the Sukhumvit area.

### 4.2. Built-Environment Characteristics: Sukhumvit District, Bangkok, Thailand

In this part, built-environment variables were assessed by cluster analysis for consideration of characteristics of the physical data. The clustering of built environments and satisfaction with QoLT based on chi-square allowed for geospatial visualization among different urban contexts. Figure 4 displays all built-environment variables by mapping via 500 × 500 grid cells. The color-temperature symbol represents the number per grid, the cool tones represent low values and the warm tones represent large values, with the unit being the number per grid, such as the number of bus stops per grid, the number of elevated train stations per grid, and the number of buildings on different grids.

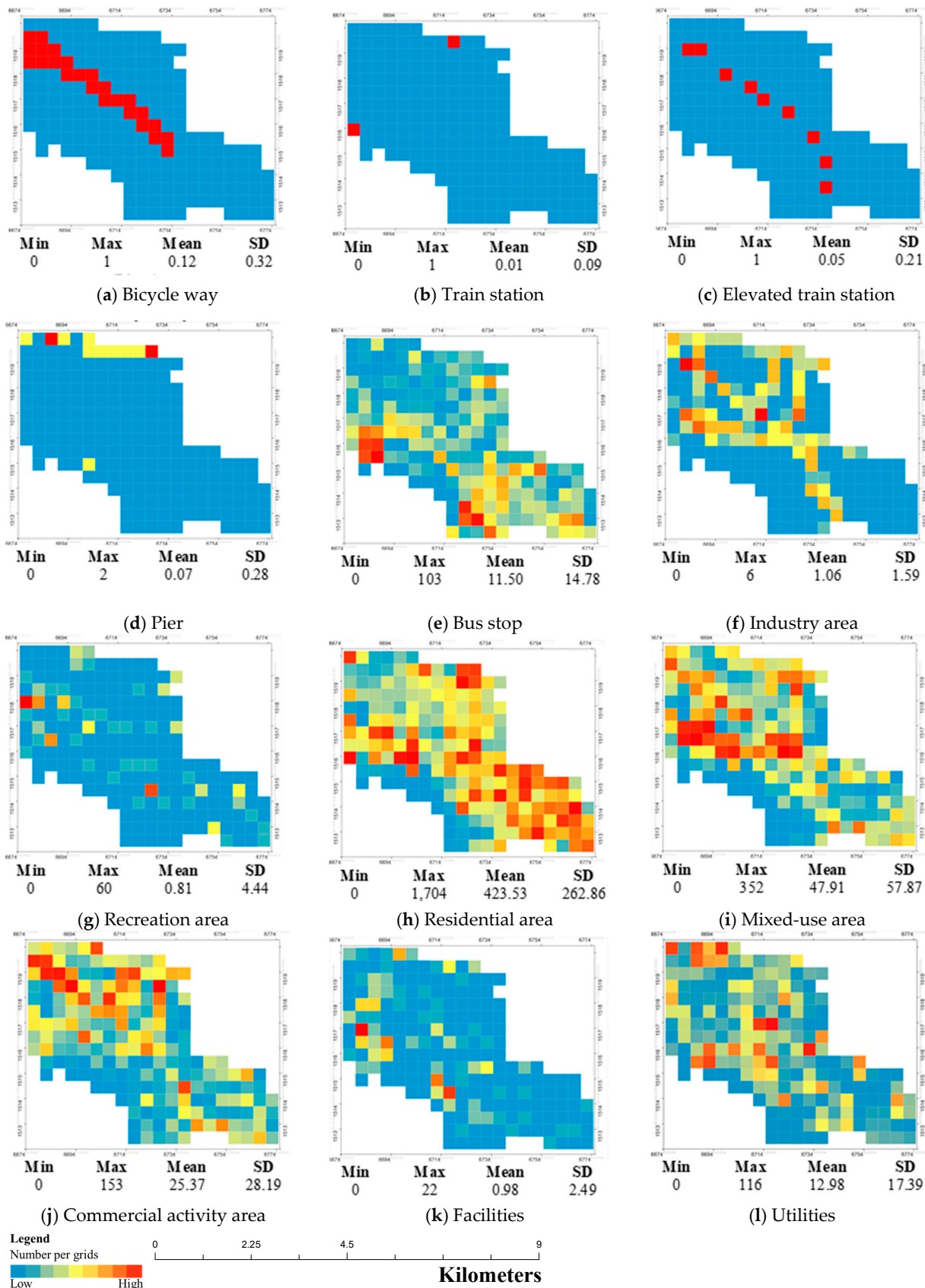

**Figure 4.** Built-environment characteristics: Sukhumvit district, Bangkok, Thailand.

The information shown in the following figure reveals that the transportation and delivery system was located mainly at the core of Sukhumvit Road, especially elevated train stations and bicycle paths. For public buses, there were service points scattered around the main roads of Klong Toei and Wattana. When considering all three districts, it was found that Phra Khanong district demonstrated fewer transportation choices and transport-system density than the Khlong Toei and Wattana districts. When considering land use, it was found that 70–80% of the building characteristics were allocated for residences, followed by commerce and mixed use, respectively. In each area, public facilities were present all over the district. In terms of semantic segmentation of roads, this research applied the technique of HRNetV2 + OCR, which could be classified into 20 sub-categories of road, sidewalk, building, wall, fence, pole, traffic light, traffic sign, vegetation, terrain, sky, person, rider, car, truck, bus, train line, motorcycle, and bicycle. However, considering only physical components (see Table 3), it could be classified into 10 sub-categories of road, sidewalk, building, wall, fence, pole, traffic light, traffic sign, vegetation, and terrain. The data reflects the different road compositions in each image, which display most of the major and minor roads, had pavements and facilities such as traffic lights and traffic signs in good condition. In contrast, most minor roads and alleys had no sidewalks due to the narrow width of the roads in the study area.

### 4.3. Satisfaction of Quality-of-Life-Related Transportation

This paper adopted seven indicators to study the satisfaction with QoLT, which consisted of accessibility, design, safety, cost, environment, mobility, and information. These indicators were measured with a five-point Likert scale, where 1 = low satisfaction and 5 = high satisfaction. Table 4 and Figure 5 illustrate the level of satisfaction of quality-of-life-related transportation. It was shown that mobility had the most average satisfaction (M = 2.76, SD = 0.92), followed by accessibility (M = 2.74, SD = 0.85), information (M = 2.74, SD = 0.91), safety (M = 2.73, SD = 0.97), cost (M = 2.66, SD = 0.89), design (M = 2.65, SD = 0.98), and environment (M = 2.44, SD = 1.07). When considering the satisfaction of QoLT and mode usage, it was found that active transportation (bicycle and pedestrian) was presented as the most average satisfaction value (M = 2.95), followed by motorcycle (M = 2.73), bus (M = 2.70), passenger car (M = 2.69), and mass transit and paratransit (M = 2.49). Based on the data analysis, the respondents were most satisfied with traveling by bicycle and on foot than by private car and rail transport, which may be due to traffic congestion and overcrowding conditions within the study area. Furthermore, using a bicycle presents a flexible mode with more convenience because the variety of activities in the study area is dense and conducive to short trips.

**Table 3.** Semantic segmentation of the road network in Sukhumvit district, Bangkok, Thailand (742 sets).

| Vadhana District | | Khlong Toei District | | Phra Khanong District | |
|---|---|---|---|---|---|
|  |  |  |  |  |  |
|  |  |  |  |  |  |
|  |  |  |  |  |  |
|  |  |  |  |  |  |

| District, % | Road | Sidewalk | Building | Wall | Fence | Pole | Traffic light | Traffic sign | Vegetation | Terrain | Sky | Person | Rider | Car | Truck | Bus | Train | Motorcycle | Bicycle |
|---|---|---|---|---|---|---|---|---|---|---|---|---|---|---|---|---|---|---|---|
| Khlong Toei | 26.86 | 1.37 | 40.39 | 0.34 | 0.7 | 0.32 | 0.01 | 0.19 | 16.13 | 0.08 | 8.63 | 1.41 | 0.05 | 2.58 | 0.1 | 0.02 | 0.06 | 0.3 | 0.05 |
| Phra Khanong | 26.16 | 1.13 | 42.94 | 0.5 | 0.68 | 0.32 | 0.09 | 0.16 | 12.57 | 0.12 | 10.6 | 1.05 | 0.04 | 3.12 | 0.07 | 0.07 | 0.12 | 0.25 | 0.01 |
| Vadhana | 24.08 | 1.59 | 46.79 | 0.41 | 0.47 | 0.32 | 0.01 | 0.17 | 12.51 | 0.05 | 8.23 | 1.53 | 0.05 | 3.14 | 0.14 | 0.08 | 0.06 | 0.34 | 0.03 |

**Table 4.** Satisfaction of QoLT and mode usage.

| Variables | Private Automobile | | | | Active Transportation | | Public Transit | | | | Para Transit | | Total | |
| | Passenger Car | | Motorcycle | | | | Bus | | Mass Rapid Transit | | | | | |
| | M | SD | M | SD | M | SD | M | SD | M | SD | M | SD | M | SD |
|---|---|---|---|---|---|---|---|---|---|---|---|---|---|---|
| Accessibility | 2.67 | 0.96 | 2.69 | 0.80 | 2.95 | 0.94 | 2.77 | 0.83 | 2.73 | 0.74 | 2.64 | 0.75 | 2.74 | 0.85 |
| Design | 2.69 | 1.06 | 2.70 | 0.95 | 2.98 | 1.05 | 2.71 | 0.96 | 2.41 | 0.92 | 2.41 | 0.82 | 2.65 | 0.98 |
| Safety | 2.76 | 0.99 | 2.86 | 0.86 | 3.06 | 1.08 | 2.75 | 1.06 | 2.47 | 0.86 | 2.46 | 0.84 | 2.73 | 0.97 |
| Cost | 2.73 | 0.95 | 2.69 | 0.87 | 2.82 | 0.90 | 2.68 | 0.93 | 2.48 | 0.83 | 2.57 | 0.81 | 2.66 | 0.89 |
| Environment | 2.58 | 1.09 | 2.59 | 0.97 | 2.92 | 1.02 | 2.45 | 1.06 | 2.01 | 1.02 | 2.05 | 1.00 | 2.44 | 1.07 |
| Mobility | 2.73 | 0.96 | 2.80 | 0.83 | 3.08 | 0.94 | 2.78 | 0.99 | 2.59 | 0.88 | 2.56 | 0.86 | 2.76 | 0.92 |
| Information | 2.64 | 0.91 | 2.77 | 0.82 | 2.84 | 0.95 | 2.75 | 0.92 | 2.71 | 0.96 | 2.76 | 0.92 | 2.74 | 0.91 |

Remark: Total data includes 500 sets.

### 4.4. Relationship between Built Environment and QoLT

For multicollinearity, it was found that the variables were highly related, and redundancy was a concern in the regression analysis. A process of checking is required for multicollinearity verification, which is applied for screening bivariate correlations and should not be higher than 0.85 [74]. The results of this analysis demonstrated the correlation among the variables for bivariate correlations, as shown in Figure 6, which indicates that the correlation between the variables was at a moderate level. The value of correlations presented a suitable value lower than 0.85 (−0.01–0.46). Thus, these variables presented the potential for estimation, as shown in Table 5.

The result of the analysis demonstrates the relationship between built environment and QoLT, in which "high satisfaction" was the reference category. For "low satisfaction," it was found that bicycle paths (*Exp(B)* = 0.416), bus stops (*Exp(B)* = 0.678), elevated train stations (*Exp(B)* = 8.462), road (*Exp(B)* = 1.049), traffic signs (*Exp(B)* = 7.326), and terrain (*Exp(B)* = 0.246) were significant. Regarding "moderate satisfaction," the results of the analysis indicate statistical significance for the availability of bicycle paths (*Exp(B)* = 0.242), bus stops (*Exp(B)* = 0.674), elevated train stations (*Exp(B)* = 15.198), utilities (*Exp(B)* = 0.889), and traffic lights (*Exp(B)* = $7.845 \times 10^{-10}$). The remarkable results of both "low satisfaction" and "moderate satisfaction" revealed interesting findings in that the odds of perceiving the proportion of bicycle paths were 0.4 times and 0.2 times higher than "high satisfaction," respectively. In other words, for a unit decrease in the proportion of bicycle paths, there was an increase of 1.5 times ($e^{0.416}$) and 1.27 times ($e^{0.242}$) in the odds of choosing "low satisfaction" and "moderate satisfaction," respectively, holding other variables at a fixed value. These findings presented the same trend for the availability of bus stops, utilities, traffic lights, and terrain. The findings suggests that transportation policy and road design should promote more green and easy access for local residents and commuters to be able to cycle within the site, which would increase the share of bicycle modes and promote better QoLT within the district. However, the availability of elevated train stations, roads, and traffic signs presented the opposite trend, which may be a result of the viewpoint that accessibility to the main modes of transportation within the site is not easy for all groups of road users.

Furthermore, for the "low satisfaction" group, more road spaces was related to a lower likelihood of bicycles. The *Exp(B)* value recommends that, when a unit of road space decreased, there was an increase of 2.86 times ($e^{1.049}$) in the odds of choosing "low satisfaction" with all other factors being equal. As for traffic signs, the findings also confirmed the same trend. However, considering the terrain of the site, it was found that the odds of this group were 0.2 times higher than "high satisfaction," and when a unit decreased in the terrain, there was an increasing of 1.28 times ($e^{0.246}$) in this group when other variables

were fixed. It was obviously demonstrated that this finding presents an important linkage to the impacts of drainage for the case of lower terrain on urban transportation in the study area, particularly for roads. The urban density together with rapid urbanization are aggravated by a lack of systematic drainage systems with regular maintenance, which is worse when considering the characteristic of the location of intensified flood-risk areas due to land subsidence. For the "moderate satisfaction" group, it was found that the odds of utilities and traffic lights presented the same trend as those of terrain in the "low satisfaction" group, where a unit decrease in the proportion of both indicators presented an increase of 2.4 times, and it was about the same for both elements of street furniture in the "moderate satisfaction" group.

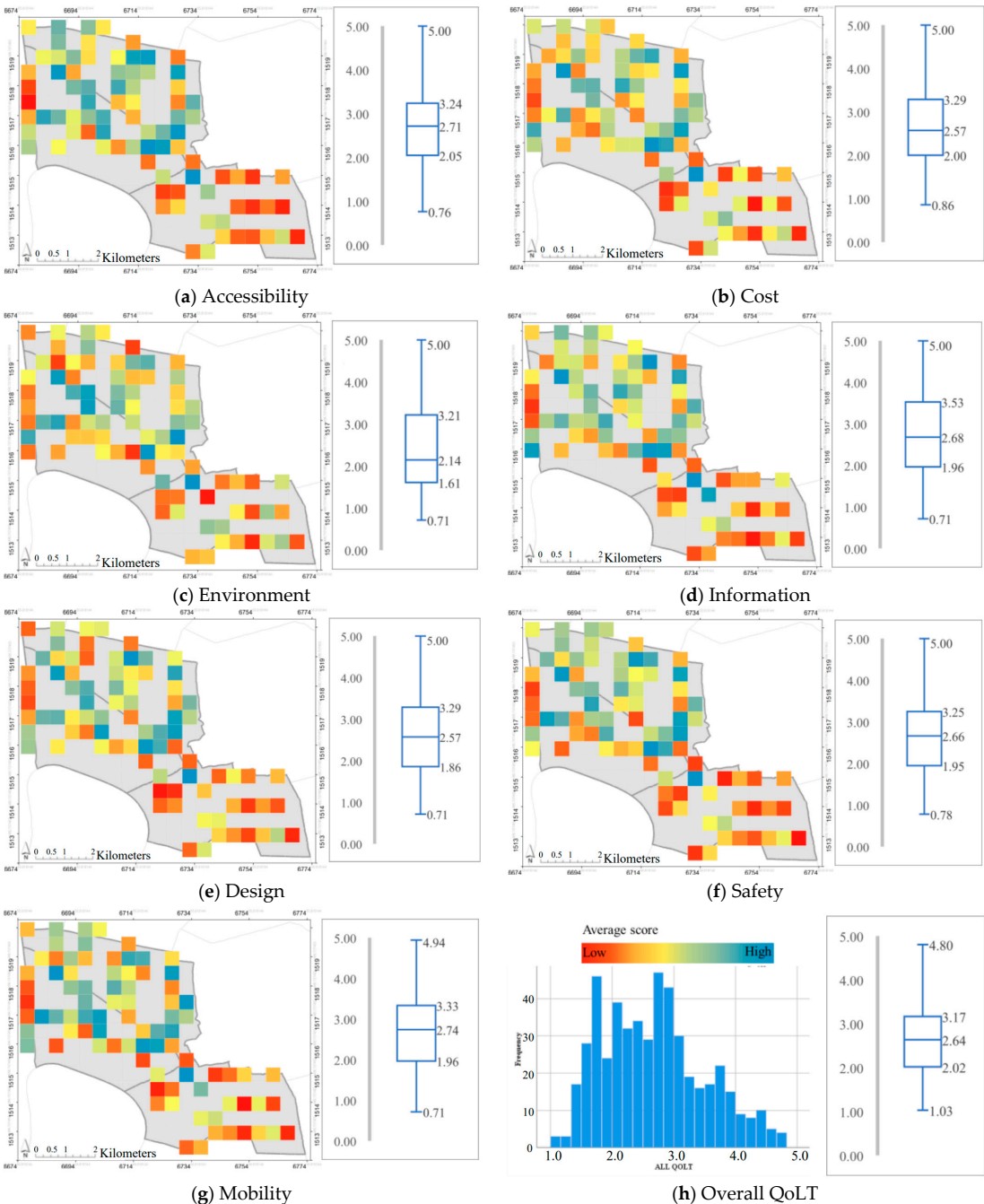

**Figure 5.** Satisfaction of QoLT: Sukhumvit district, Bangkok, Thailand.

| | Pier | Bicycle way | Bus stop | Elevated train station | Subway station | Residential | Commercial | Mixed-use | Industry | Facility | Utility | Recreation | Road | Sidewalk | Building | Wall | Fence | Pole | traffic light | traffic sign | Vegetation | Terrain |
|---|---|---|---|---|---|---|---|---|---|---|---|---|---|---|---|---|---|---|---|---|---|---|
| Pier of water transport | | −0.09 * | 0.14 ** | −0.07 | 0.42 ** | −0.19 ** | −0.01 | −0.08 | −0.06 | 0.29 ** | −0.01 | −0.02 | −0.04 | −0.06 | 0.15 ** | −0.06 | −0.06 | −0.04 | −0.03 | −0.04 | −0.13 ** | −0.01 |
| Bicycle way | −0.09 * | | 0.31 ** | 0.46 ** | 0.09 | −0.18 ** | 0.33 ** | 0.35 ** | −0.19 ** | −0.09 | 0.05 | −0.03 | 0.10 * | 0.00 | 0.16 ** | 0.02 | −0.08 | −0.08 | −0.05 | 0.03 | −0.17 ** | 0.04 |
| Bus stop | 0.14 ** | 0.31 ** | | 0.45 ** | 0.12 ** | −0.15 ** | 0.35 ** | 0.37 ** | −0.16 ** | 0.27 ** | 0.12 * | −0.12 ** | 0.03 | 0.00 | 0.18 ** | −0.02 | −0.05 | −0.03 | −0.09 * | 0.05 | −0.20 ** | −0.08 |
| Elevated train station | −0.07 | 0.46 ** | 0.45 ** | | 0.14 ** | −0.19 ** | 0.32 ** | 0.12 ** | −0.03 | 0.14 ** | 0.11 * | −0.04 | 0.17 ** | 0.04 | 0.11 * | −0.02 | 0.00 | −0.05 | −0.04 | 0.12 * | −0.18 ** | −0.04 |
| Subway station | 0.42 ** | 0.09 | 0.12 ** | 0.14 ** | | −0.16 ** | 0.14 ** | −0.05 | −0.04 | 0.09 * | −0.01 | −0.03 | 0.09 | −0.02 | 0.06 | −0.01 | −0.06 | −0.03 | −0.02 | 0.00 | −0.11 * | −0.03 |
| Residential | −0.19 ** | −0.18 ** | −0.15 ** | −0.19 ** | −0.16 ** | | −0.22 ** | 0.37 ** | 0.26 ** | −0.11 * | −0.10 * | −0.21 ** | −0.10 * | −0.21 ** | −0.01 | 0.06 | 0.19 ** | −0.02 | 0.05 | 0.13 ** | 0.05 | 0.07 |
| Commercial | −0.01 | 0.33 ** | 0.35 ** | 0.32 ** | 0.14 ** | −0.22 ** | | 0.29 ** | −0.26 ** | −0.04 | 0.03 | −0.10 * | −0.02 | 0.12 * | 0.15 ** | −0.07 | −0.05 | −0.07 | −0.03 | 0.08 | −0.17 ** | −0.05 |
| Mixed-use | −0.08 | 0.35 ** | 0.37 ** | 0.12 ** | −0.05 | 0.37 ** | 0.29 ** | | −0.06 | −0.06 | 0.23 ** | −0.12 ** | −0.07 | −0.08 | 0.13 ** | −0.03 | −0.02 | −0.06 | −0.07 | 0.06 | −0.10 * | −0.06 |
| Industry | −0.06 | −0.19 ** | −0.16 ** | −0.03 | −0.04 | 0.26 ** | −0.26 ** | −0.06 | | −0.17 ** | 0.01 | −0.10 * | −0.04 | −0.08 | −0.02 | 0.04 | 0.06 | 0.12 ** | 0.05 | 0.03 | −0.02 | 0.05 |
| Facility | 0.29 ** | −0.09 | 0.27 ** | 0.14 ** | 0.09 * | −0.10 * | −0.04 | −0.06 | −0.17 ** | | 0.18 ** | 0.11 * | 0.13 ** | 0.03 | −0.01 | −0.01 | 0.00 | 0.07 | −0.05 | 0.05 | −0.12 ** | −0.02 |
| Utility | −0.01 | 0.05 | 0.12 * | 0.10 * | −0.01 | −0.10 * | 0.03 | 0.23 ** | 0.01 | 0.18 ** | | −0.01 | 0.05 | −0.01 | 0.07 | −0.07 | 0.00 | 0.00 | −0.04 | −0.03 | −0.12 ** | −0.01 |
| Recreation | −0.02 | −0.03 | −0.12 ** | −0.05 | −0.03 | −0.21 ** | −0.10 * | −0.12 ** | −0.10 * | 0.11 * | −0.01 | | 0.02 | 0.21 ** | −0.13 ** | −0.03 | 0.01 | 0.08 | 0.01 | −0.05 | 0.06 | 0.01 |
| Road | −0.04 | 0.10 * | 0.03 | 0.17 ** | 0.09 | −0.11 * | −0.02 | −0.07 | −0.04 | 0.13 ** | 0.05 | 0.02 | | 0.07 | −0.47 ** | −0.03 | −0.01 | 0.03 | 0.05 | −0.03 | −0.08 | 0.06 |
| Sidewalk | −0.06 | 0.01 | −0.01 | 0.04 | −0.02 | −0.21 ** | 0.12 * | −0.08 | −0.08 | 0.03 | −0.01 | 0.21 ** | 0.07 | | −0.25 ** | 0.13 ** | 0.13 ** | 0.41 ** | −0.09 * | 0.04 | 0.17 ** | −0.07 |
| Building | 0.15 ** | 0.16 ** | 0.18 ** | 0.11 * | 0.06 | −0.01 | 0.15 ** | 0.13 ** | −0.02 | −0.01 | 0.07 | −0.13 ** | −0.47 ** | −0.25 ** | | −0.21 ** | −0.24 ** | −0.34 ** | −0.06 | −0.09 * | 0.14 ** | −0.07 |
| Wall | −0.06 | 0.02 | −0.02 | −0.02 | −0.01 | 0.06 | −0.07 | −0.03 | 0.04 | −0.01 | −0.07 | −0.03 | −0.03 | 0.13 ** | −0.21 ** | | 0.11 * | 0.24 ** | 0.10 * | 0.06 | 0.14 ** | 0.01 |
| Fence | −0.06 | −0.08 | −0.05 | −0.01 | −0.06 | 0.19 ** | −0.05 | −0.02 | 0.06 | 0.00 | 0.00 | 0.01 | −0.01 | 0.13 ** | −0.24 ** | 0.11 * | | 0.27 ** | −0.02 | 0.07 | 0.18 ** | 0.08 |
| Pole | −0.04 | −0.08 | −0.03 | −0.05 | −0.03 | −0.02 | −0.07 | −0.06 | 0.12 ** | 0.07 | 0.00 | 0.08 | 0.03 | 0.41 ** | −0.34 ** | 0.24 ** | 0.27 ** | | −0.03 | 0.17 ** | 0.20 ** | 0.05 |
| Traffic light | −0.03 | −0.05 | −0.09 * | −0.04 | −0.03 | 0.05 | −0.03 | −0.07 | 0.05 | −0.05 | −0.04 | 0.01 | 0.05 | −0.091 * | −0.06 | 0.10 * | −0.02 | −0.03 | | 0.19 ** | −0.05 | −0.02 |
| Traffic sign | −0.04 | 0.03 | 0.05 | 0.115 * | 0.00 | 0.13 ** | 0.08 | 0.06 | 0.03 | 0.05 | −0.03 | −0.05 | −0.03 | 0.04 | −0.09 * | 0.06 | 0.07 | 0.17 ** | 0.19 ** | | 0.06 | 0.07 |
| Vegetation | −0.13 ** | −0.17 ** | −0.20 ** | −0.182 ** | −0.11 * | 0.05 | −0.17 ** | −0.10 * | −0.02 | −0.08 | −0.12 ** | 0.06 | −0.08 | 0.17 ** | −0.66 ** | 0.14 ** | 0.18 ** | 0.20 ** | −0.05 | 0.06 | | −0.02 |
| Terrain | −0.01 | 0.04 | −0.08 | −0.04 | −0.03 | 0.07 | −0.05 | −0.06 | 0.05 | −0.02 | −0.01 | 0.01 | 0.06 | −0.07 | −0.07 | 0.01 | 0.08 | 0.05 | −0.02 | 0.07 | −0.02 | |

Remark:　　−1 ▢▢▢▢▢ 1　Correlation

**Figure 6.** Correlation of independent variables. Note: * = Tests were evaluated at the 0.05 level of statistical significance; ** = tests were evaluated at the 0.01 level of statistical significance.

**Table 5.** Relationship between built-environment attributes and QoLT.

| Main Variable | Sub-Variables | Low Satisfaction | | | | | Moderate Satisfaction | | | | |
|---|---|---|---|---|---|---|---|---|---|---|---|
| | | $B$ | Std. Error | Wald | Exp($B$) | Sig. | $B$ | Std. Error | Wald | Exp($B$) | Sig. |
| Mode of transportation (GIS) | Pier for water transport | 0.382 | 0.612 | 0.390 | 1.466 | 0.532 | 0.753 | 0.641 | 1.382 | 2.124 | 0.240 |
| | Bicycle path | −0.876 | 0.445 | 3.885 | 0.416 | 0.049 * | −1.418 | 0.511 | 7.713 | 0.242 | 0.005 * |
| | Bus stop | −0.389 | 0.101 | 14.859 | 0.678 | 0.000 ** | −0.395 | 0.111 | 12.624 | 0.674 | 0.000 ** |
| | Elevated train station | 2.136 | 0.689 | 9.605 | 8.462 | 0.002 * | 2.721 | 0.725 | 14.089 | 15.198 | 0.000 ** |
| | Subway station | −0.374 | 0.859 | 0.190 | 0.688 | 0.663 | −0.701 | 0.924 | 0.576 | 0.496 | 0.448 |
| Land-use characteristic (GIS) | Residential | 0.002 | 0.001 | 3.082 | 1.002 | 0.079 | .001 | 0.001 | 0.241 | 1.001 | 0.624 |
| | Commercial | −0.004 | 0.005 | 0.684 | 0.996 | 0.408 | −0.007 | 0.005 | 1.774 | 0.993 | 0.183 |
| | Mixed use | −0.002 | 0.003 | 0.456 | 0.998 | 0.500 | .003 | 0.004 | .840 | 1.003 | 0.359 |
| | Industry | 0.004 | 0.016 | 0.076 | 1.004 | 0.782 | −0.021 | 0.017 | 1.486 | 0.979 | 0.223 |
| | Facility | 0.008 | 0.010 | 0.644 | 1.008 | 0.422 | −0.006 | 0.011 | .261 | 0.994 | 0.610 |
| | Utility | −0.051 | 0.047 | 1.150 | 0.950 | 0.283 | −0.118 | 0.057 | 4.226 | 0.889 | 0.040 * |
| | Recreation | 0.044 | 0.049 | 0.809 | 1.045 | 0.368 | −0.011 | 0.062 | 0.030 | 0.989 | 0.862 |
| Road components (semantic segmentation) | Road | 0.048 | 0.024 | 3.921 | 10.049 | 0.048 * | 0.031 | 0.026 | 1.493 | 1.032 | 0.222 |
| | Sidewalk | −0.074 | 0.113 | 0.430 | 00.928 | 0.512 | −0.119 | 0.123 | 0.933 | 0.888 | 0.334 |
| | Building | −0.016 | 0.020 | 0.617 | 00.984 | 0.432 | −0.011 | 0.021 | 0.263 | 0.989 | 0.608 |
| | Wall | 0.091 | 0.169 | 0.291 | 1.095 | 0.590 | 0.151 | 0.173 | 0.763 | 1.163 | 0.382 |
| | Fence | 0.039 | 0.121 | 0.102 | 1.040 | 0.750 | 0.094 | 0.130 | 0.521 | 1.099 | 0.470 |
| | Pole | 0.523 | 0.609 | 0.736 | 1.687 | 0.391 | 0.648 | 0.651 | 0.990 | 1.912 | 0.320 |
| | Traffic light | −0.619 | 0.887 | 0.487 | 0.538 | 0.485 | −20.966 | 10.005 | 4.391 | $7.845 \times 10^{-10}$ | 0.036 * |
| | Traffic sign | 1.991 | 0.965 | 4.255 | 7.326 | 0.039 * | 1.764 | 0.998 | 3.122 | 5.836 | 0.077 |
| | Vegetation | −0.046 | 0.024 | 3.677 | 0.955 | 0.055 | −0.038 | 0.025 | 2.369 | 0.962 | 0.124 |
| | Terrain | −1.404 | 0.611 | 5.275 | 0.246 | 0.022 * | −1.084 | 0.654 | 2.751 | 0.338 | 0.097 |
| −2 Log likelihood | 814.767 | | | | | | | | | | |
| Chi-square | 111.551 | | | | | | | | | | |
| Significant | 0.000 | | | | | | | | | | |
| McFadden | 0.115 | | | | | | | | | | |
| Percent correct predicted | 59.0 | | | | | | | | | | |

Note: The reference category is high satisfaction; * = tests were evaluated at the 0.05 level of statistical significance; ** = tests were evaluated at the 0.01 level of statistical significance.

The results interestingly point out which built environment affects QoLT based on satisfaction through the respondents' perceptions. Several built-environment variables in terms of transportation modes and road components presented the most significant variables of QoLT. The details of these components were bicycle paths, bus stops, elevated train stations, utilities, road space, traffic lights, traffic signs, and terrain, which make up the main structure of the transportation system.

*4.5. Cluster of Built Environment and Semantic Segmentation of Road Components*

Considering the general information of the built environment as mentioned above, such data were taken into statistical consideration to cluster them so that each group's

particularities differed according to the physical characteristics. A statistical analysis of the clustering was performed, and the results are shown in Table 6.

It was found that a total of three groups could be categorized based on the property of their different physical characteristics. The details are as follows:

*Cluster 1* is a group of densely populated residential areas or slums located in the Khlong Toei area. In addition to the density of residential areas, there are no pedestrian walkways. However, this area is close to various transportation and transportation systems. There is also a variety of both commercial and mixed-use activities.

*Cluster 2* comprises commercial and business areas with moderately dense housing. Most activities are commercial activities close to various transportation and transport systems.

*Cluster 3* presents a residential area mixed with commercial and small industrial activities. Most of them are in the Phra Khanong area, which has fewer transportation and transport systems than clusters 1 and 2.

**Table 6.** ANOVA of the clusters.

| Main Variable (Data-Gathering Techniques) | Sub-Variables | Mean Square | F | Sig. |
|---|---|---|---|---|
| Mode of transportation (GIS) | Pier | 0.246 | 3.149 | 0.044 * |
| | Bicycle path | 1.48 | 10.987 | 0.000 ** |
| | Bus stop | 32.397 | 10.2 | 0.000 ** |
| | Railway station | 0.048 | 4.1 | 0.017 * |
| | Elevated train station | 1.282 | 13.976 | 0.000 ** |
| | Subway station | 0.017 | 495 | 0.61 |
| Land-use characteristic (GIS) | Residential | 11,928,595.98 | 821.219 | 0.000 ** |
| | Commercial | 21,586.794 | 22.008 | 0.000 ** |
| | Mixed use | 74,679.659 | 20.189 | 0.000 ** |
| | Industry | 2971.661 | 25.945 | 0.000 ** |
| | Facility | 2859.988 | 9.72 | 0.000 ** |
| | Utility | 31.674 | 3.848 | 0.022 * |
| | Recreation | 108.628 | 2.523 | 0.081 |
| Road components (semantic segmentation) | Road | 133.423 | 1.626 | 0.198 |
| | Sidewalk | 18.103 | 8.53 | 0.000 ** |
| | Building | 366.696 | 1.815 | 0.164 |
| | Wall | 2.433 | 1.789 | 0.168 |
| | Fence | 11.663 | 5.3 | 0.005 ** |
| | Pole | 0.04 | 0.405 | 0.667 |
| | Traffic light | 0.164 | 3.131 | 0.045 * |
| | Traffic sign | 0.646 | 9.5 | 0.000 ** |
| | Vegetation | 3.683 | 0.033 | 0.967 |
| | Terrain | 0.066 | 1.46 | 0.233 |

The grouping of areas mentioned above reflects that the characteristics of each group are different in terms of land use and transportation systems. Therefore, this section considers the association between satisfaction with QoLT among different clusters (see Figure 7). Table 7 illustrates significant satisfaction with QoLT in terms of accessibility, safety, and cost dimensions, and these associations could explain the differences among the three

clusters. The findings are consistent with the physical characteristics of each group in terms of differences in access to transport systems, including the safety and cost of transport in the commercial area of cluster 2 dominating others. This is because cluster 1 is a group in which most of the sample was in a crowded residential area with limited economic conditions and access to safe travel patterns compared to the areas in cluster 2 and cluster 3.

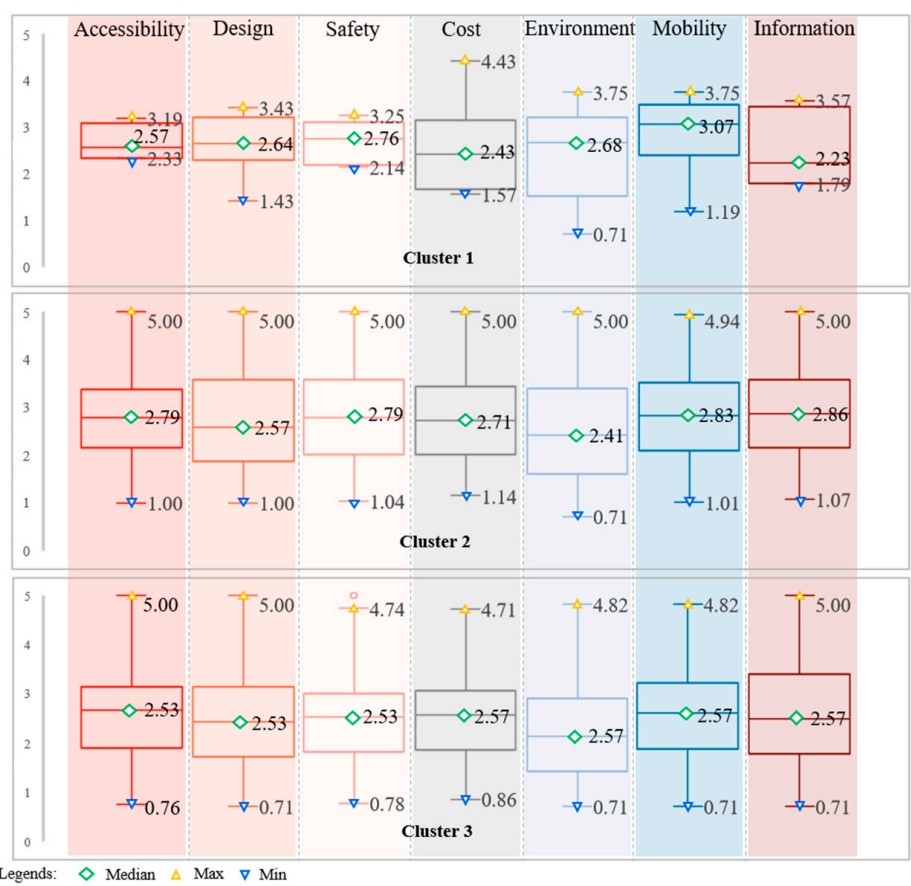

**Figure 7.** Satisfaction with QoLT classified by clustering of Sukhumvit district, Bangkok, Thailand.

**Table 7.** Association of satisfaction with QoLT among different clusters.

| | Variables | Cluster 1 | Cluster 2 | Cluster 3 | Total | *p*-Value |
|---|---|---|---|---|---|---|
| Accessibility | Very dissatisfied | 0.0 | 7.1 | 21.0 | 13.0 | 0.000 ** |
| | Dissatisfied | 50.0 | 32.1 | 28.0 | 30.6 | |
| | Medium | 50.0 | 36.4 | 37.9 | 37.2 | |
| | Satisfied | 0.0 | 15.7 | 9.8 | 13.0 | |
| | Very satisfied | 0.0 | 8.6 | 3.3 | 6.2 | |
| | Mean | 2.67 | 2.86 | 2.57 | 2.74 | |
| Design | Very dissatisfied | 16.7 | 19.6 | 26.2 | 22.4 | 0.298 |
| | Dissatisfied | 33.3 | 31.1 | 27.6 | 29.6 | |
| | Medium | 33.3 | 21.8 | 27.6 | 24.4 | |
| | Satisfied | 16.7 | 16.1 | 11.7 | 14.2 | |
| | Very satisfied | 0.0 | 11.4 | 7.0 | 9.4 | |
| | Mean | 2.64 | 2.75 | 2.51 | 2.65 | |

**Table 7.** *Cont.*

| Variables | | Cluster 1 | Cluster 2 | Cluster 3 | Total | *p*-Value |
|---|---|---|---|---|---|---|
| Safety | Very dissatisfied | 0.0 | 15.4 | 22.4 | 18.2 | 0.004 ** |
| | Dissatisfied | 33.3 | 28.9 | 31.3 | 30.0 | |
| | Medium | 66.7 | 27.5 | 32.7 | 30.2 | |
| | Satisfied | 0.0 | 15.4 | 8.9 | 12.4 | |
| | Very satisfied | 0.0 | 12.9 | 4.7 | 9.2 | |
| | Mean | 2.69 | 2.89 | 2.52 | 2.73 | |
| Cost | Very dissatisfied | 33.3 | 13.2 | 21.5 | 17.0 | 0.012 * |
| | Dissatisfied | 33.3 | 36.1 | 30.8 | 33.8 | |
| | Medium | 16.7 | 25.0 | 32.2 | 28.0 | |
| | Satisfied | 0.0 | 17.1 | 12.6 | 15.0 | |
| | Very satisfied | 16.7 | 8.6 | 2.8 | 6.2 | |
| | Mean | 2.55 | 2.76 | 2.53 | 2.66 | |
| Environment | Very dissatisfied | 33.3 | 36.4 | 43.0 | 39.2 | 0.073 |
| | Dissatisfied | 16.7 | 17.1 | 19.6 | 18.2 | |
| | Medium | 33.3 | 21.8 | 24.3 | 23.0 | |
| | Satisfied | 16.7 | 14.3 | 10.3 | 12.6 | |
| | Very satisfied | 0.0 | 10.4 | 2.8 | 7.0 | |
| | Mean | 2.44 | 2.57 | 2.26 | 2.44 | |
| Mobility | Very dissatisfied | 16.7 | 15.7 | 22.0 | 18.4 | 0.073 |
| | Dissatisfied | 0.0 | 26.1 | 26.2 | 25.8 | |
| | Medium | 66.7 | 31.4 | 33.6 | 32.8 | |
| | Satisfied | 16.7 | 15.7 | 14.0 | 15.0 | |
| | Very satisfied | 0.0 | 11.1 | 4.2 | 8.0 | |
| | Mean | 2.88 | 2.87 | 2.60 | 2.76 | |
| Information | Very dissatisfied | 33.3 | 14.6 | 25.2 | 19.4 | 0.060 |
| | Dissatisfied | 33.3 | 24.3 | 26.2 | 25.2 | |
| | Medium | 16.7 | 34.6 | 25.2 | 30.4 | |
| | Satisfied | 16.7 | 16.8 | 17.8 | 17.2 | |
| | Very satisfied | 0.0 | 9.6 | 5.6 | 7.8 | |
| | Mean | 2.50 | 2.85 | 2.61 | 2.74 | |
| Number of cases in each cluster | | 6 (1.20%) | 280 (56.00%) | 214 (42.80%) | 500 (100%) | |

Remark: Total data includes 500 sets; * = tests were evaluated at the 0.05 level of statistical significance; ** = tests were evaluated at the 0.01 level of statistical significance.

## 5. Discussion

Exploring the spatial effects of built environments on QoLT based on an integration of GIS and deep-learning approaches was the main objective of this study. This study attempted to understand QoLT by applying a new approach by adopting the technology of GIS and deep-learning techniques to provide a better understanding of physical data attributes. This is due to studies of the built environment being constantly concerned with one or several aspects related to quality of life, such as physical activities, mobility, public health, etc. [75]. A number of studies has pointed to the link between the built environment and quality of life, with the accessibility of transport nodes presenting a key

role in transportation research. However, fewer studies have pointed to an association between the built environment and transportation-related quality of life.

The results revealed that the subjective quality or perception of individuals of the built-environment characteristics allows the quality of the urban environment's influence on commuters to be described. This basic association between individuals and quality of life refers to the social and economic characteristics of people who also play an important role in quality of life. The social and economic characteristics present explanatory variables to express the position or role in society, needs, perceptions, and so on. Importantly, social and economic characteristics create divisions of social class from different social and individual capital. These relate to accessing different opportunities for creating a better life and well-being [76,77]. The socio-economic characteristics of this study also reflect the diversity of people, especially the issue of income disparities, where high-income people tend to travel by private car. This is reflected in the individual satisfaction from different groups of people in distinct areas or environments that demonstrate variations in the perceived quality of life. This was confirmed with the result of clustering analysis in this study among cluster 1, cluster 2, and cluster 3. That is to say, individuals may be attracted to different places based on their preferences.

For example, those who highly value transit may be more likely to live in places with a quality transit system. This was confirmed by several studies that attempted to demonstrate the importance of built environments on quality-of-life-related transportation. For example, Mattson et al. (2017) found that those who had recently missed a trip because of a lack of transportation or who reported more significant difficulties in making trips reported lower overall life satisfaction after controlling for other factors such as age and health [78]. Likewise, Kong (2017) studied the relationship between QoL and the built environment by considering categories of built-environment variables that included transit accessibility, population density, and mixed uses of residential, commercial, and industrial spaces for a positive relationship with QoL [79]. In concordance with this result, the importance of transport access and land-use factors was reflected, which can explain differences in built-environment characteristics, especially for land use related to mixed uses of residential, commercial, and industrial spaces. Planners and researchers consider accessibility a critical strategy to maximize environmental sustainability and QoL in urban areas [80].

Moreover, when considering the relationships of different variables of built environments and QoLT, it was shown that a number of categories of built-environment variables (e.g., bicycle path, bus stop, elevated train station, utility, road, traffic light, and signage) are significant predictors of life satisfaction in terms of QoLT. Importantly, this research applied combined methodologies of a traditional approach via a survey of respondents' perceptions after assessing subjective aspects and a technological approach for evaluating the tangible environmental characteristics. The clustering of urban typology-based GIS applications merging environmental data from deep-learning technique can help by incorporating more dimensions of built-environment datasets. Although the present research focused only on a specific area of CBD in Bangkok, with the powerful tool of the data-generating approach, the comprehensive view of urban development by covering the Bangkok Metropolitan Region (BMR) can be recognized to promote more choices of transport mode while creating environmentally sustainable urban development. Finally, transportation plans or improvements will enable all stakeholders to understand one of the key indicators used to measure the quality of life of people who are living in urban areas. Notably, transport-related built-environment characteristics involve multiple variables that are influenced by various geographical and social constraints, such as proximity to the facility and population density [81,82]. This study adapted quality-of-life indicators related to sentimental transport systems. Data were collected based on people's feelings regarding environmental factors related to transport systems instead of utilizing only physical data. Therefore, for transportation planning to lead to a better quality of life, those factors need to be tested to become more accurate and meet urban- and transportation-development needs.

In addition to, fewer studies have pointed to an association between the built environment and transportation-related quality of life (QoLT). Importantly, in this research, there was no semantic label on the images of Bangkok used as the input due to the characteristics of the area and physical road components collected in the study area being more diverse than those in other countries. However, the images' semantic output was checked for the quality of data and it was found that it was generally correct, although less accurate than those from Europe cases. Therefore, the challenge of this study was applying indicators of physical components to determine quality of life in Thailand that could provide useful results to generate new findings. For future studies, in order to gain a better understanding of quality-of-life assessment in travel, it is necessary to adopt new techniques and tools due to social-dimension constraints of the emergence of new tools such as smartphones, GPS tracking, machine learning, deep learning, etc. Each tool provides different benefits; thus, the application of analysis tools must be taken into account for the suitability of the data and to stay within the scope of the study.

## 6. Conclusions

This study explored the spatial aspects of quality of life related to transportation by integrating GIS and deep-learning techniques. Sukhumvit district, Bangkok, Thailand, was targeted as a case study, which comprises three districts: Vadhana district, Khlong Toei district, and Phra Khanong district. QoLT-related data were collected from 500 respondents. Data collection was conducted by multiple sources: Geographic Information System (GIS), information extracted using the semantic-segmentation technique, and a questionnaire survey. The findings demonstrates that a number of categories of built-environment variables were significant predictors of life satisfaction in terms of QoLT, and this integrated procedure allowed for a better understanding of how different urban patterns result in different QoLT among different groups of road users. Finally, this study highlighted essential aspects of urban-planning and transport systems that must consider differences not only in terms of the physical characteristics of the site, but also in terms of social and economic characteristics of the people living there. This will help improve the quality of travel in everyday life, which creates a balance at the social, economic, and environmental levels for sustainable development.

**Author Contributions:** Conceptualization, P.I.; Methodology, P.I., B.K. and Y.I.; Formal analysis, P.I., S.C. and P.K.; Investigation, S.C.; Data curation, S.C.; Writing—original draft, P.I. and S.C.; Writing—review & editing, P.I., S.C., Y.H., B.K. and Y.I.; Visualization, S.C.; Supervision, P.I., Y.H., B.K. and Y.I. All authors have read and agreed to the published version of the manuscript.

**Funding:** This research was funded by Japan International Cooperation Agency.

**Institutional Review Board Statement:** The study was conducted according to the guidelines of the Declaration of Helsinki and approved from the Human Research Ethics Committee of Thammasat University Social Sciences (certificate of approval number 146/2021, 9 June 2022).

**Informed Consent Statement:** Informed consent was obtained from all subjects involved in the study.

**Data Availability Statement:** Not applicable.

**Acknowledgments:** The authors gratefully acknowledge the support provided by JICA in "The Project of Smart Transport Strategy for THAILAND 4.0". The authors also thank Chayapol Parisonyu, Natchanon Vorayuth and Nattaphum Jampalee (Department of Computer Engineering, Faculty of Engineering, Chulalongkorn University, Bangkok, Thailand) for assistance with implementing the image-recognition software. The research was conducted by the Center of Excellence in Urban Mobility Research and Innovation (UMRI), Thammasat University, Pathumthani, Thailand.

**Conflicts of Interest:** The authors declare no conflict of interest.

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
