# Peer review of "Exploring the Spatial Effects of Built Environment on Quality of Life Related Transportation by Integrating GIS and Deep Learning Approaches"

_sustainability, doi:10.3390/su15032785_

Round 1

Reviewer 1 Report

In this paper, the quality of life-related to transportation is studied by a combination method of GIS, DDR-Net, and questionnaire survey. It can be concluded from the results the social and economic characteristics of people also play an important role in the quality of life-related to transportation, some conclusions can be used for reference to the balanced development of urban areas.

If this paper is finally published in this journal, the following amendments may help to improve the manuscript.

1.      The title of the paper is deep learning, but the paper only uses the existing mature model as a tool, and still uses traditional analysis methods in the main analysis parts such as relevance analysis.

2.      In fact, this paper uses three compound means to obtain data. The deep learning method is only used as a tool to obtain actual street information, rather than the main innovation or content of this paper. It is suggested to consider the title of the paper.

3.      The paper uses DDR-Net as a tool for target recognition, but does not describe the selection of super parameters, training times, training results, recognition success rate, etc. In the process of deep learning recognition, it can be seen from the title of the paper that the author takes deep learning as an important content of this chapter, and suggests to supplement the details of deep learning.

4.      In this paper, several abbreviations such as "QOL", "QoL", etc., are suggested to be unified.

Author Response

Exploring the Spatial Effects of Built Environment on Quality of Life Related Transportation by Integrating GIS and Deep Learning Approaches

Response to Reviewer No.1

In this paper, the quality of life-related to transportation is studied by a combination method of GIS, DDR-Net, and questionnaire survey. It can be concluded from the results the social and economic characteristics of people also play an important role in the quality of life-related to transportation, some conclusions can be used for reference to the balanced development of urban areas.

If this paper is finally published in this journal, the following amendments may help to improve the manuscript.

Point 1:      The title of the paper is deep learning, but the paper only uses the existing mature model as a tool, and still uses traditional analysis methods in the main analysis parts such as relevance analysis.

Response1: We are aware of your concern about the application of deep learning methods. Thus, we are modifying the scope of the “Literature review” and “methods” that explain the relationship of deep learning with semantic segmentation and Convolutional neural network (CNN) which represent as methods in the main analysis of our research.

The detail of revision is presented in section 2 and section 3, page 4 which are “Section 2: Literature review. Quality of life (QoL) can be classified as objective domain (Sirgy et al., 2006), subjective domain (Diener, 2000), combination between objective and subjective (Ferkany, 2012), and domain specific (Drobnič et al., 2010; Furie and Desai, 2012) … CNN prunes the edges using two mechanisms, convolutional layers, and subsampling (or Max-pooling) layers.” And “Section 3: Methodology. This study aims to explore the spatial effects of built environments on quality of life-related transportation through an integrating of questionnaire survey for collecting subjective data based on life satisfaction among six indicators (accessibility, design, safety, cost, environment, mobility, information) … The package includes the pre-trained model; hence, the software is ready to use, and no need to re-train the model.”

Point 2:      In fact, this paper uses three compound means to obtain data. The deep learning method is only used as a tool to obtain actual street information, rather than the main innovation or content of this paper. It is suggested to consider the title of the paper.

Response2: We are glad about your comments on suggestion to consider the title of the paper. However, we attempt adding the scope of analysis, literature review and result of data analysis to point to relationship of deep learning with semantic segmentation and Convolutional neural network (CNN) which are methods in the main analysis our research. The detail of revision was performed in Section 1 (Introduction, page 3), Section 2 (Literature review, page 4-6), Section3 (Methodology, page 6-9).

Point 3:     The paper uses DDR-Net as a tool for target recognition, but does not describe the selection of super parameters, training times, training results, recognition success rate, etc. In the process of deep learning recognition, it can be seen from the title of the paper that the author takes deep learning as an important content of this chapter, and suggests to supplement the details of deep learning.

Response3: We agreed with your comment that we are about missing the detail of tool for target recognition. However, this research uses HRNetV2 + OCR as a tool for target recognition. According to the additional contents, it was revised as shown in Page 5, lines 211, the software package is pre-trained by the authors and ready to use. As we have already chosen the OCRNet as the model, the only super parameter is "backbone=HRNet_W48". (Edit in page 8, Methodology part).

Point 4:      In this paper, several abbreviations such as "QOL", "QoL", etc., are suggested to be unified.

Response4: We agreed with your comments, we edited "QoL" as to be unified in whole paper.

Reviewer 2 Report

This paper studies the relationship between built environment and Quality of Life Related Transportation (QoLT) using GIS and questionnaire data. My main concern is that the methodology is a bit outdated in this area of research. In the past decade, with the emergence of smartphone and IoT technologies, the most recent research uses tracking data to map the participants’ activities with the built environment for detailed analysis. One particularly important issue is the mode of transport, as the authors discussed in their paper. It will be difficult to map the specific mode of transport to the built environment using a questionnaire.

I understand the experiment is already finished, but I suggest you review and summarise relevant papers to demonstrate your understanding of state of the art. For example

-          Zhao, P., Haitao, H., Li, A., & Mansourian, A. (2021). Impact of data processing on deriving micro-mobility patterns from vehicle availability data. Transportation Research Part D: Transport and Environment, 97, 102913.

-          Wang, F., & Chen, C. (2018). On data processing required to derive mobility patterns from passively-generated mobile phone data. Transportation Research Part C: Emerging Technologies, 87, 58-74.

-          Zeitler, E., Buys, L., Aird, R., & Miller, E. (2012). Mobility and active ageing in suburban environments: Findings from in-depth interviews and person-based GPS tracking. Current Gerontology and Geriatrics Research, 2012.

Some other comments and questions:

-          The flow between methodologies is not clear in Figure 3. Why are GIS and Built Environment both pointing to GIS analysis? Aren’t they the same thing? I also do not follow what is the difference between multi-method and multi-approach?

-          The quality of life will be directly influenced by social-economic status. I do not see how this major factor is accounted.

-          How did you apply ‘Deep Dual-resolution Networks’? Is it through some software package or did you code this yourself?

-          How did you map modes of transport to the built environment? Or are they treated as independent variables? (obviously they have a strong correlation)

Author Response

Exploring the Spatial Effects of Built Environment on Quality of Life Related Transportation by Integrating GIS and Deep Learning Approaches

Response to Reviewer No.2

This paper studies the relationship between built environment and Quality of Life Related Transportation (QoLT) using GIS and questionnaire data.

Point 1:      My main concern is that the methodology is a bit outdated in this area of research. In the past decade, with the emergence of smartphone and IoT technologies, the most recent research uses tracking data to map the participants’ activities with the built environment for detailed analysis. One particularly important issue is the mode of transport, as the authors discussed in their paper. It will be difficult to map the specific mode of transport to the built environment using a questionnaire.

Response1: We are aware of your concern and the revision was made at the introduction part (Page 3) as “Well known that in several studies pointed to the quality of life would be directly influenced by social-economic status…. Besides, several studies mainly applied questionnaires or physical survey approaches in-built environment-related spatial dimensions.”. For the mode of transportation, it can be drawn from the questionnaire survey data which was reflected by individual level at the specific location of spatial data and the image data.

The detail of methodology was modified for clearer explaining about this point as seen in Page 6-9. The modified part is as “This study aims to explore the spatial effects of built environments on quality of life-related transportation through an integrating of questionnaire survey for collecting subjective data based on life satisfaction among six indicators (accessibility, design, safety, cost, environment, mobility, information)… The model is implemented in Python and trained by using the cityscape dataset. The package includes the pre-trained model; hence, the software is ready to use, and no need to re-train the model.”.

Point 2:      I understand the experiment is already finished, but I suggest you review and summarise relevant papers to demonstrate your understanding of state of the art. For example

-          Zhao, P., Haitao, H., Li, A., & Mansourian, A. (2021). Impact of data processing on deriving micro-mobility patterns from vehicle availability data. Transportation Research Part D: Transport and Environment, 97, 102913.

-          Wang, F., & Chen, C. (2018). On data processing required to derive mobility patterns from passively-generated mobile phone data. Transportation Research Part C: Emerging Technologies, 87, 58-74.

-          Zeitler, E., Buys, L., Aird, R., & Miller, E. (2012). Mobility and active ageing in suburban environments: Findings from in-depth interviews and person-based GPS tracking. Current Gerontology and Geriatrics Research, 2012.

Response2: We are aware of your concern and added issue in limitations of this research as shown in the Introduction part (Page 3) and the Discussion part (Page 18-19). The detail of the revision is “Notably, due to fewer studies have pointed to an association between the built environment and transportation-related quality of life (QoLT). Importantly, in this research, there is no semantic label on the Bangkok image use as the input, due to the characteristics of the area and road physical components collected in the study area are more di-verse than those in other countries…. Each tool provides different benefits, thus the application of analysis tools must be taken into account for the suitability of the data and within the scope of the study.”

Point 3: The flow between methodologies is not clear in Figure 3. Why are GIS and Built Environment both pointing to GIS analysis? Aren’t they the same thing? I also do not follow what is the difference between multi-method and multi-approach?

Response3: We are glad about your comments on detail of the methodology. Thus, we edited The detail of methodology was modified for clearer explaining about this point as seen in Page 6-9, particularly Fig3 was also revised according to your comments.

Point 4:       The quality of life will be directly influenced by social-economic status. I do not see how this major factor is accounted.

Response4: We are glad about your comments however, this study focuses only two dimensions which are built environments and satisfaction on quality-of-life related transportation. Because the study of social and economic factors affecting quality of life is well known as the main factor affecting quality of life and is widely studied.

However, in this study, we attempted to scope of our study by explaining in Section 4 “the descriptive analysis together with multinomial logistic model was ap-plied to provide better understanding about the different levels of QoLT based on model estimation. Furthermore, the clustering model is employed to understand the different urban context related to a variation level of their QoLT. Finally, the summary of the re-sult of analysis was concluded and the policy recommendation are discussed in the last sections (section 5 and section 6).”

Furthermore, the results of analysis in Section 4 also provided 4.1 The information of Socio-economic participant’s profile and 4.2 Built environment characteristic: Sukhumvit district, Bangkok, Thailand.

Point 5:       How did you apply ‘Deep Dual-resolution Networks’? Is it through some software package or did you code this yourself?

Response 5: This research uses HRNetV2 + OCR as a tool for target recognition, furthermore, the open-source software package of "PaggleSeg" (2021) was also applied in the semantic segmentation process, published at PaddlePaddle Contributors (2019). The model used in this research is OCRNet, and the selected backbone is HRNet_W48, the only super parameter is "backbone=HRNet_W48". The model is implemented in Python and trained using the cityscape dataset. The package includes the pre-trained model; hence, the software is ready to use, and no need to re-train the model. The detail of revision is demonstrated in the Methodology part, Page 8.

Point 6:      How did you map modes of transport to the built environment? Or are they treated as independent variables? (obviously they have a strong correlation)

Response6: We are glad about your comments on the correlation between modes of transport to the built environment. Thus, we added correlate consideration in Fig.6 (Page 13, the Results part.

Reviewer 3 Report

Although an interesting research study has been done by the authors, there are vague points in this research.

1-      What do you mean by this sentence? “Quality of Life Related Transportation” What are the quality indicators in this study?

2- What variables have been investigated and what are the effects of these variables?

3- In conclusion, the spatial effects on the quality of transportation have not been determined precisely?

4- Research method and conclusions are not stated in the abstract?

5- What is the reason for choosing the geographical area of the study? And what are the characteristics of this place, the studied people, etc.?

6- Parameters such as travel time, travel quality, travel cost, etc. have not been investigated. Is there a special reason?

7- Do the investigated variables have an effect on choosing the studied transportation modes?

Author Response

Exploring the Spatial Effects of Built Environment on Quality of Life Related Transportation by Integrating GIS and Deep Learning Approaches

Response to Reviewer No.3

Although an interesting research study has been done by the authors, there are vague points in this research.

Point 1: What do you mean by this sentence? “Quality of Life Related Transportation” What are the quality indicators in this study?

Response1: Quality of Life Related Transportation is one of the specific domains in quality. Several studies was reviewed as shown in Table 1 for the several indicators in investigating “Quality of Life Related Transportation” such as accessibility, safety, cost, etc. (Page 3-4, Literature review part)

Point 2: What variables have been investigated and what are the effects of these variables?

Response2: Two domain variables were investigated which are quality of life related transportation (life satisfaction) and built environment. All variables are shown in Table 1. For effects of these variables, we apply multinomial regression analysis that can show significant in positive and negative of independent (built environment) with dependent (quality of life related transportation) variable at a 0.05 level of statistical significance. The results is demonstrated in Section 4 (Results).

Point 3:  In conclusion, the spatial effects on the quality of transportation have not been determined precisely?

Response3: We are glad about your comments. We added this issue in conclusion (Section 6, Page 19). The detail is  “Findings demonstrates a number of built environment variable categories are significant predictor of life satisfaction in term of QoLT and this integrating procedure allows better understanding of how different urban patterns resulted to different QoLT among differ-ent group of road users…... It will definitely help improving the quality of traveling in everyday life which creates a balance at the social, economic, and environmental levels for sustainable development.”

Point 4:  Research method and conclusions are not stated in the abstract?

Response 4: We are glad about your comments. Thus, we edited our abstract in Page 1, (Abstract part). The detail is “This study explores the spatial effects of built environment on quality of life related to transporta-tion (QoLT) through the  combination of GIS application and deep learning based on questionnaire survey by focusing on a case study in Sukhumvit district, Bangkok, Thailand…. The better understanding of QoLT creates an important value for transportation development to balance at the social, economic, and environmental levels toward sustainable futures.”.

Point 5:  What is the reason for choosing the geographical area of the study? And what are the characteristics of this place, the studied people, etc.?

Response 5: We are aware of your concern, so we added the reason for choosing the geographical area of the study in Page 7 (Section1, Methodology part). The detail is “Sukhumvit district, Bangkok, Thailand, is presented as a case study which consists of three districts (Vadhana district, Khlong Toei district, and Phra Khanong district) as depicted in Fig. 2… This area is considered as an significant representative of urban problems that exert a certain degree of pressure on their inhabitants to search for desire a certain quality of life according to diversity of spatial urban and facility characteristics.”.

Point 6:   Parameters such as travel time, travel quality, travel cost, etc. have not been investigated. Is there a special reason?

Response 6: In this study, the physical components of roads and cities were focused as the social factors, so these factors (travel time, travel quality, travel cost) were not taken into consideration. However, the point that you suggest is an interesting point that further studies need to consider, in which point we add it to the limits of the study as shown in the Discussion section (Page 18), the detail is “Notably, due to fewer studies have pointed to an association between the built environment and transportation-related quality of life (QoLT)… Each tool provides different benefits, thus the application of analysis tools must be taken into account for the suitability of the data and within the scope of the study.”

Point 7:    Do the investigated variables have an effect on choosing the studied transportation modes?

Response 7: This aims to explore the spatial effects of built environments on quality of life-related transportation based on an integrating of GIS and deep learning techniques. The transportation modes were included in the built environment variables in term of supply side as explained in Section 4.5 (Page 16). The detail of analysis is “It was found that a total of three groups could be categorized by the property of their different physical characteristics. The details are as follows…. The findings are consistent with the physical characteristics of each group with differences in access to transport systems, including the safety and cost of transport in commercial area of cluster 2 is dominating others. This is because cluster 1 is a group where most of the sample is in a crowded residential area with limited economic conditions and access to safe travel patterns compared to areas in cluster 2 and cluster 3.”.

Round 2

Reviewer 1 Report

In this paper, the quality of life-related to transportation is studied by a combination method of GIS, DDR-Net, and questionnaire survey. It can be concluded from the results the social and economic characteristics of people also play an important role in the quality of life-related to transportation, some conclusions can be used for reference to the balanced development of urban areas.

The author has made point-to-point changes to the comments made last time, I think the current paper can be published in this journal.

Author Response

Exploring the Spatial Effects of Built Environment on Quality of Life Related Transportation by Integrating GIS and Deep Learning Approaches

Reviewer 1

Round II

Point 1:

In this paper, the quality of life-related to transportation is studied by a combination method of GIS, DDR-Net, and questionnaire survey. It can be concluded from the results the social and economic characteristics of people also play an important role in the quality of life-related to transportation, some conclusions can be used for reference to the balanced development of urban areas.

Response: We agreed with your comments and we made the additional discussion on page 18, Discussion section.

Reviewer 2 Report

The authors have made some revisions to clarify the methodology. As a result, I can now understand it better, but I still have some major questions about your methodology.

-          What is the resolution of your grid in Fig4? It seems very coarse, could you justify this?

-          What are the units in Fig4? For example, with a,b,c, it is either red or blue. Is it binary? And with f,g,h,I, what does the ‘number of areas’ mean? Do you mean the square meters area value? All others need to be clarified too.

-          It should be noted that, with transport-related built environment characteristics, it is not just the absolute value that matters, but the proximity to these facilities (e.g. train stations, bus stops). See the following paper on how GIS is used to analyse transport.

Mangold, M., Zhao, P., Haitao, H., & Mansourian, A. (2022). Geo-fence planning for dockless bike-sharing systems: a GIS-based multi-criteria decision analysis framework. Urban Informatics, 1(1), 1-15.

-          Your semantic segmentation, where does the ‘296 road segments’ come from? How are they selected and sampled? Why are the particular picture in Figure 3 used? Why are they representative (of a very large area)?

-          I still do not follow how your questionnaire can be mapped. E.g. is there any relationship between Table 4 and Fig5 which are discussed together? They seem to me unrelated studies with no logic to link them.

-          I notice you added some highlighted literature on page 3, but they do not appear in the reference list.

Author Response

Exploring the Spatial Effects of Built Environment on Quality of Life Related Transportation by Integrating GIS and Deep Learning Approaches

Reviewer 2

Round II

Point 1: What is the resolution of your grid in Fig4? It seems very coarse, could you justify this?

Response: We are aware of your concern. Thus, we added some justify about resolution of grid on page 8-9, Methodology part.

Point 2: What are the units in Fig4? For example, with a,b,c, it is either red or blue. Is it binary? And with f,g,h,I, what does the ‘number of areas’ mean? Do you mean the square meters area value? All others need to be clarified too.

Response: We are aware of your concern. Thus, we added some details for describe units of variable, on Page 11, Results section.

Point3: It should be noted that, with transport-related built environment characteristics, it is not just the absolute value that matters, but the proximity to these facilities (e.g. train stations, bus stops). See the following paper on how GIS is used to analyse transport.

Mangold, M., Zhao, P., Haitao, H., & Mansourian, A. (2022). Geo-fence planning for dockless bike-sharing systems: a GIS-based multi-criteria decision analysis framework. Urban Informatics, 1(1), 1-15.

Response: We agreed with your comments and the additional discussion has been modified as shown in Page 23, Discussion section.

Point4: Your semantic segmentation, where does the ‘296 road segments’ come from? How are they selected and sampled? Why are the particular picture in Figure 3 used? Why are they representative (of a very large area)?

Response: We added a description of how to get of 742 road segments on Page 8, Methodology section. For the particular picture in Figure 3, this picture presents as a sample picture shown by the researcher to show an overview of the consideration from the picture.

Point5:  I still do not follow how your questionnaire can be mapped. E.g., is there any relationship between Table 4 and Fig5 which are discussed together? They seem to me unrelated studies with no logic to link them.

Response: We agree with your comments. We added how our questionnaire can be mapped on page 8, Methodology section. This will help to understand the connection between Figure 5 and Table 4, which is the same data but shows different results. Table 4 presents the mean value of the statistical indicators while Figure 5 presents the values of analysis. The metrics are displayed via an area of grid to visualize the distribution of quality of life indicators.

Point 6:  I notice you added some highlighted literature on page 3, but they do not appear in the reference list.

Response: We checked all references throughout the manuscript and added some missing references as suggested.

Round 3

Reviewer 2 Report

The authors clarified my questions.